# Integrative analysis of genomic and transcriptomic characteristics associated with progression of aggressive thyroid cancer

Seong-Keun Yoo[1,2,3,14], Young Shin Song [ORCID] [4,5,14], Eun Kyung Lee [ORCID] [6], Jinha Hwang[7], Hwan Hee Kim[4], Gyeongseo Jung[4], Young A Kim[8], Su-jin Kim[9], Sun Wook Cho[4], Jae-Kyung Won[10], Eun-Jae Chung[11], Jong-Yeon Shin[1,3], Kyu Eun Lee[1,9], Jong-Il Kim [ORCID] [1,7], Young Joo Park [ORCID] [1,4,15] & Jeong-Sun Seo[1,3,7,12,13,15]

Anaplastic thyroid cancer (ATC) and advanced differentiated thyroid cancers (DTCs) show fatal outcomes, unlike DTCs. Here, we demonstrate mutational landscape of 27 ATCs and 86 advanced DTCs by massively-parallel DNA sequencing, and transcriptome of 13 ATCs and 12 advanced DTCs were profiled by RNA sequencing. *TERT*, *AKT1*, *PIK3CA*, and *EIF1AX* were frequently co-mutated with driver genes (*BRAF*^V600E and *RAS*) in advanced DTCs as well as ATC, but tumor suppressors (e.g., *TP53* and *CDKN2A*) were predominantly altered in ATC. *CDKN2A* loss was significantly associated with poor disease-specific survival in patients with ATC or advanced DTCs, and up-regulation of *CD274* (PD-L1) and *PDCD1LG2* (PD-L2). Transcriptome analysis revealed a fourth molecular subtype of thyroid cancer (TC), ATC-like, which hardly reflects the molecular signatures in DTC. Furthermore, the activation of JAK-STAT signaling pathway could be a potential druggable target in *RAS*-positive ATC. Our findings provide insights for precision medicine in patients with advanced TCs.

[1] Genomic Medicine Institute, Medical Research Center, Seoul National University, Seoul 03080, Republic of Korea. [2] Interdisciplinary Program in Bioinformatics, Seoul National University, Seoul 08826, Republic of Korea. [3] Macrogen Inc., Seoul 08511, Republic of Korea. [4] Department of Internal Medicine, Seoul National University College of Medicine, Seoul 03080, Republic of Korea. [5] Department of Internal Medicine, CHA Bundang Medical Center, CHA University, Seongnam 13496, Republic of Korea. [6] Center for Thyroid Cancer, National Cancer Center, Goyang 10408, Republic of Korea. [7] Department of Biomedical Sciences, Seoul National University Graduate School, Seoul 03080, Republic of Korea. [8] Department of Pathology, Seoul National University Boramae Medical Center, Seoul 07061, Republic of Korea. [9] Department of Surgery, Seoul National University College of Medicine, Seoul 03080, Republic of Korea. [10] Department of Pathology, Seoul National University College of Medicine, Seoul 03080, Republic of Korea. [11] Department of Otorhinolaryngology-Head and Neck Surgery, Seoul National University College of Medicine, Seoul 03080, Republic of Korea. [12] Precision Medicine Center, Seoul National University Bundang Hospital, Seongnam, Bundang-gu, Gyeonggi-do 13605, Republic of Korea. [13] Gong-Wu Genomic Medicine Institute, Seoul National University Bundang Hospital, Seongnam 13605, Republic of Korea. [14] These authors contributed equally: Seong-Keun Yoo, Young Shin Song. [15] These authors jointly supervised this work: Young Joo Park, Jeong-Sun Seo. Correspondence and requests for materials should be addressed to Y.J.P. (email: yjparkmd@snu.ac.kr) or to J.-S.S. (email: jeongsunseo@gmail.com)

The molecular understating of differentiated thyroid cancer (DTC) was expanded by recent comprehensive studies[1,2]. The Cancer Genome Atlas (TCGA) proposed two molecular subtypes of papillary thyroid cancer (PTC), $BRAF^{V600E}$-like and $RAS$-like, based on transcriptome analysis[1]. Moreover, our group demonstrated a third molecular subtype, Non-$BRAF$-Non-$RAS$ (NBNR), which is closely associated with follicular-patterned thyroid tumors, including follicular adenoma (FA) and minimally invasive follicular thyroid cancer (miFTC)[2]. According to the aforementioned studies, the molecular classification of thyroid cancer (TC) better explains its underlying characteristics than a histological classification.

Anaplastic TC (ATC), which accounts for 2% or fewer of TC cases, is one of the most aggressive human malignancies[3]. The median survival of ATC patients is about 3–5 months after diagnosis[4]. In addition, advanced DTCs, such as poorly differentiated, metastatic, or widely invasive types, also show poor outcomes[5–7]. Nonetheless, there is no effective therapy to prolong the survival of patients with those forms of TC. Although the molecular characteristics of DTC have been analyzed, the underlying mechanism of its progression to advanced DTC and ATC has not been fully elucidated. Several studies have reported that multiple mutational hits in tumor suppressor genes (TSGs) or oncogenes were involved in the development of ATC, but the majority of those reports were confined to genomic alterations[8,9]. Thus, the need for further transcriptomic analysis of ATC and advanced DTCs is increased to discover molecular mechanisms potentially involved in tumor progression and targets for treatment.

In this study, we apply various types of massively parallel sequencing technology to 113 advanced TCs, including 27 ATCs and 86 advanced DTCs, to reveal their genomic and transcriptomic characteristics. We expect that this work will broaden the current molecular understanding of advanced TCs and lead to more efficient diagnostic and therapeutic strategies for them.

## Results

**Mutational landscape of ATC and advanced DTCs.** We have preliminarily analyzed 13 ATCs, 3 focal ATC/poorly differentiated TCs (PDTCs), and 9 widely invasive follicular TCs (wiFTCs) by whole-genome sequencing (WGS) or whole-exome sequencing (WES), and extended the dataset with 88 additional samples using targeted sequencing. In total, 113 advanced TCs, including 27 ATCs, 15 PDTCs, 28 focal ATC/PDTCs, 12 wiFTCs, and 31 metastatic papillary TCs (PTCs) were investigated for mutational profiling. Targeted sequencing was also performed on 13 ATCs and 3 focal ATC/PDTCs which were analyzed by WGS and the concordance rate between two methods was 91.89% (Supplementary Data 1). We collected tissues from primary (76/113), distant metastatic (19/113), and locally recurred or residual sites (18/113), respectively. The clinicopathological characteristics of the patients according to histology are shown in Table 1.

The mutational landscape of 113 advanced TCs is illustrated in Fig. 1. $BRAF^{V600E}$ and $RAS$ (40.74% and 44.44%, respectively) were recognized as major driver genes in ATC, whereas no fusion gene was identified. $RET$ fusions ($CCDC6$-$RET$ and $NCOA4$-$RET$) were discovered in PDTC, focal ATC/PDTC, and metastatic PTC. We also found $NFE2L2$ mutation which is frequently altered in lung squamous cell carcinoma and recently identified in TC as fusion driver[10,11]. In PDTC, $BRAF^{V600E}$ and $RAS$ mutations were found with the same frequency (26.67%), but most focal ATC/PDTCs had a $BRAF^{V600E}$ mutation (82.14%). wiFTC and metastatic PTC showed frequencies of $BRAF^{V600E}$ (0% and 64.52%, respectively) and $RAS$ mutations (66.67% and 22.58%, respectively) similar to those found in FA/miFTC and PTC,

respectively[1,2]. In metastatic PTC, we also discovered $SPOP^{P94R}$ which is repeatedly reported in various types of TC[2,12,13].

$TERT$ was the most recurrently co-mutated gene in ATC, with a frequency of 55.56% (Fig. 2a), as previously reported[9] (56.12%; Supplementary Fig. 1a). It was also frequently altered in advanced DTCs (46.67%, 39.29%, and 47.39% in PDTCs, focal ATC/PDTCs, and metastatic PTCs, respectively). Notably, wiFTC showed an extremely high frequency of $TERT$ alterations (91.67%), even more frequent than in ATC, and other alterations were barely discovered in this subtype. We also identified $ATRX$ mutations in ATC, focal ATC/PDTC, wiFTC, and metastatic PTC.

Furthermore, 74.07% of ATCs harbored mutations in TSGs such as $TP53$, $CDKN2A$, $PTEN$, $LATS1$, $LATS2$, $CTNNA2$, $TET1$, and $BRCA1$ (Fig. 2b). Among them, $TP53$ was the most commonly altered TSG (48.15%; Fig. 2c). Notably, we confirmed two types of germline $TP53$ mutation (E11Q and R49H) in ATC (11.11%), wiFTC (8.33%), and metastatic PTC (3.23%; Supplementary Fig. 2). TSG mutations were also found in advanced DTCs (26.67%, 21.43%, 25.81%, and 16.67% in PDTCs, focal ATC/PDTCs, metastatic PTCs, and wiFTCs, respectively), and these frequencies were much higher than those of DTC[1,2] (1.41% in PTCs and 7.41% in FA/miFTCs).

We also identified $AKT1$/$PIK3CA$ and $EIF1AX$ co-mutations in ATC (Fig. 2d), as in previous reports[8,9]. Co-mutations of these genes were frequent in ATCs with $BRAF^{V600E}$ (36.36%) and $RAS$ (66.67%) mutations, respectively, and they were mutually exclusive with each other (Fig. 3e). $AKT1$/$PIK3CA$ and $EIF1AX$ mutations also showed increased incidences with the aggressiveness. Additionally, a somatic mutation in $U2AF1$ (S34F), which is a spliceosomal machinery gene known to be involved in the progression of acute myeloid leukemia[14], was identified in one focal ATC/PDTC (Fig. 1).

The additional mutated genes, such as $KMT2D$, $ATM$, $CHEK2$, $ATM$, $CHEK2$, $NF1$, $NF2$, and $MEN1$, that are putatively involved in the progression of TC were also discovered (Supplementary Figs. 1b and 3). However, it is possible that they might be rare germline mutations as matched normal DNA of the targeted sequenced tumors were not analyzed.

**Somatic $TERT$ rearrangements in wiFTC.** As we described earlier, $TERT$ alteration was most frequently identified in wiFTC (91.67%). In this subtype, two structural rearrangements within or adjacent to $TERT$, as well as promoter mutations, were found. We identified $TERT$ fusion gene, $PDE8B$-$TERT$, which has not been described in TC (Fig. 2f). After the breakpoint, elevated $TERT$ expression was found which is a hallmark of fusion gene[15] (Fig. 2g). $PDE8B$ is known to be involved in thyroid function and actively expressed in the thyroid gland[16,17]. Although TCGA group did not describe $TERT$ fusion in TC, another group identified $MTMR12$-$TERT$ fusion in PTC (TCGA-BJ-A4O9-01) from TCGA cohort (see URLs). We found remarkably elevated $TERT$ expression in wiFTC with $PDE8B$-$TERT$ and PTC with $MTMR12$-$TERT$ from TCGA (Fig. 2h and Supplementary Fig. 4).

Meanwhile, one wiFTC showed increased expression of $TERT$ without fusion or promoter mutations, hence we performed WGS to identify a structural rearrangement adjacent to $TERT$ as previous reports[18,19]. As a result, an inter-chromosomal translocation, t(2;5)(2q;5p), at 21 kilobases upstream from $TERT$ was discovered (Supplementary Fig. 5). The huge up-regulation of $TERT$ in tumors with intergenic rearrangements is known to be induced by super-enhancer hijacking[18,19]. We also pinpointed typical-enhancers and super-enhancer in the partner region of upstream translocation (Fig. 2i). The distance between hijacked super-enhancer and $TERT$ (179 kb) was closer than that of the original target gene of the super-enhancer, $RGPD4$ (337 kb).

**Table 1 Clinicopathological characteristics of patients according to the histology**

|  |  | ATC | PDTC | Focal ATC/PDTC | wiFTC | Metastatic PTC |
|---|---|---|---|---|---|---|
| N |  | 27 | 15 | 28 | 12 | 31 |
|  | Fresh frozen tissue | 14 | 2 | 5 | 9 | 4 |
|  | FFPE tissue | 13 | 13 | 23 | 3 | 27 |
| Age[a] |  |  |  |  |  |  |
|  | Initial diagnosis | 60.2 ± 15.3 | 55.3 ± 19.9 | 56.0 ± 15.9 | 65.3 ± 12.7 | 56.0 ± 9.7 |
|  | Surgery[b] | 64.7 ± 12.4 | 55.8 ± 20.1 | 57.9 ± 16.5 | 65.4 ± 12.7 | 60.4 ± 8.2 |
| Male, n (%) |  | 10 (37.0) | 4 (26.7) | 8 (28.6) | 6 (50.0) | 9 (29.0) |
| Tumor origin |  |  |  |  |  |  |
|  | PTC, n (%) | 16 (59.3) | 6 (40.0) | 27 (96.4) | 0 (0) | 31 (100) |
|  | FTC, n (%) | 8 (29.6) | 6 (40.0) | 1 (3.6) | 12 (100) | 0 (0) |
|  | Unknown, n (%) | 3 (11.1) | 3 (20.0) | 0 (0) | 0 (0) | 0 (0) |
| Distant metastasis, n (%) |  | 20 (74.1) | 6 (40.0) | 7 (25.0) | 10 (83.3) | 31 (100) |
| Final disease status |  |  |  |  |  |  |
|  | NED, n (%) | 3 (11.1) | 11 (73.3) | 19 (67.9) | 3 (25) | 3 (9.7) |
|  | AWD, n (%) | 4 (14.8) | 2 (13.3) | 4 (14.3) | 8 (66.7) | 16 (51.6) |
|  | DOD, n (%) | 20 (74.1) | 2 (13.3) | 5 (17.9) | 1 (8.3) | 12 (38.7) |
| Disease-specific survival, months[c] |  | 6.9 (2.4–13.2) | 60.3 (36.9–121.0) | 109.2 (20.1–124.7) | 25.7 (13.6–73.7) | 44 (21.8–109.8) |

*ATC* anaplastic thyroid cancer, *PDTC* poorly differentiated thyroid cancer, *wiFTC* widely invasive follicular thyroid cancer, *PTC* papillary thyroid cancer, *FTC* follicular thyroid cancer, *NED* no evidence of disease, *FFPE* formalin-fixed paraffin-embedded, *AWD* alive with disease, *DOD* death of disease
[a]Values presented as mean ± standard deviation
[b]Age at surgery for analyzed tissue
[c]Values presented as median (interquartile range)

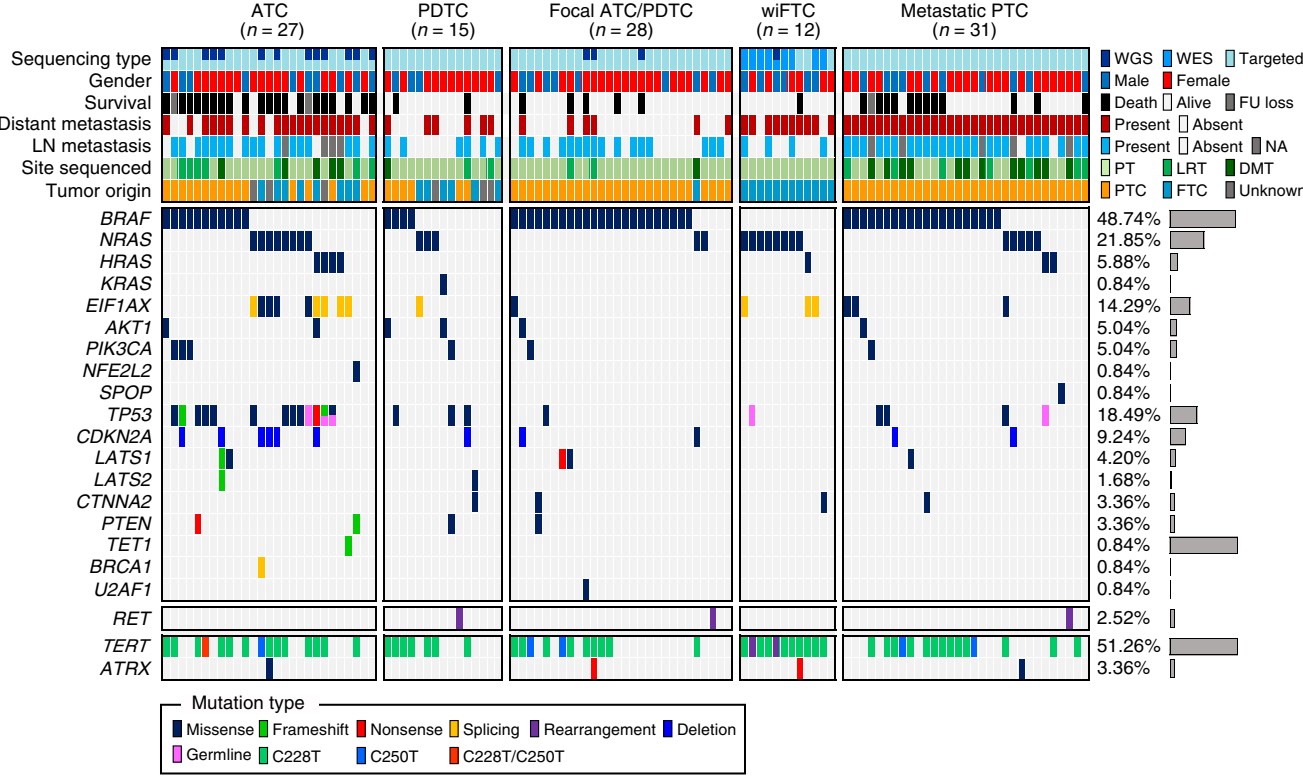

**Fig. 1** The mutational landscape of ATC and advanced DTCs. Each column represents an individual tumor. Only genes harboring mutations confirmed to be somatic in at least one tumor were displayed. Genes were sorted by their functions as oncogenes, tumor suppressors, splicing machinery gene, *RET* fusion, and telomere lengthening genes. The right bar chart represents the frequencies of gene alterations across 113 advanced TCs. FU, LN, NA, PT, LRT, and DMT indicate follow-up, lymph node, not available, primary tumor, locally recurred or residual tumor, and distant metastatic tumor, respectively

Moreover, we found the unusual intergenic expression by RNA sequencing (RNA-seq) alignment after the breakpoint, which suggests the consequential effect of this translocation (Supplementary Fig. 6). Two *TERT* rearrangements were confirmed by polymerase chain reaction (PCR) and Sanger sequencing

(Supplementary Fig. 7). Additionally, the expression levels of *TERT*, induced by promoter mutations, were significantly higher in ATC than in DTC (*P* = 0.006; Fig. 2h). This may be due to the expansion of sub-clones with *TERT* promoter mutations in ATC, as previously described[8].

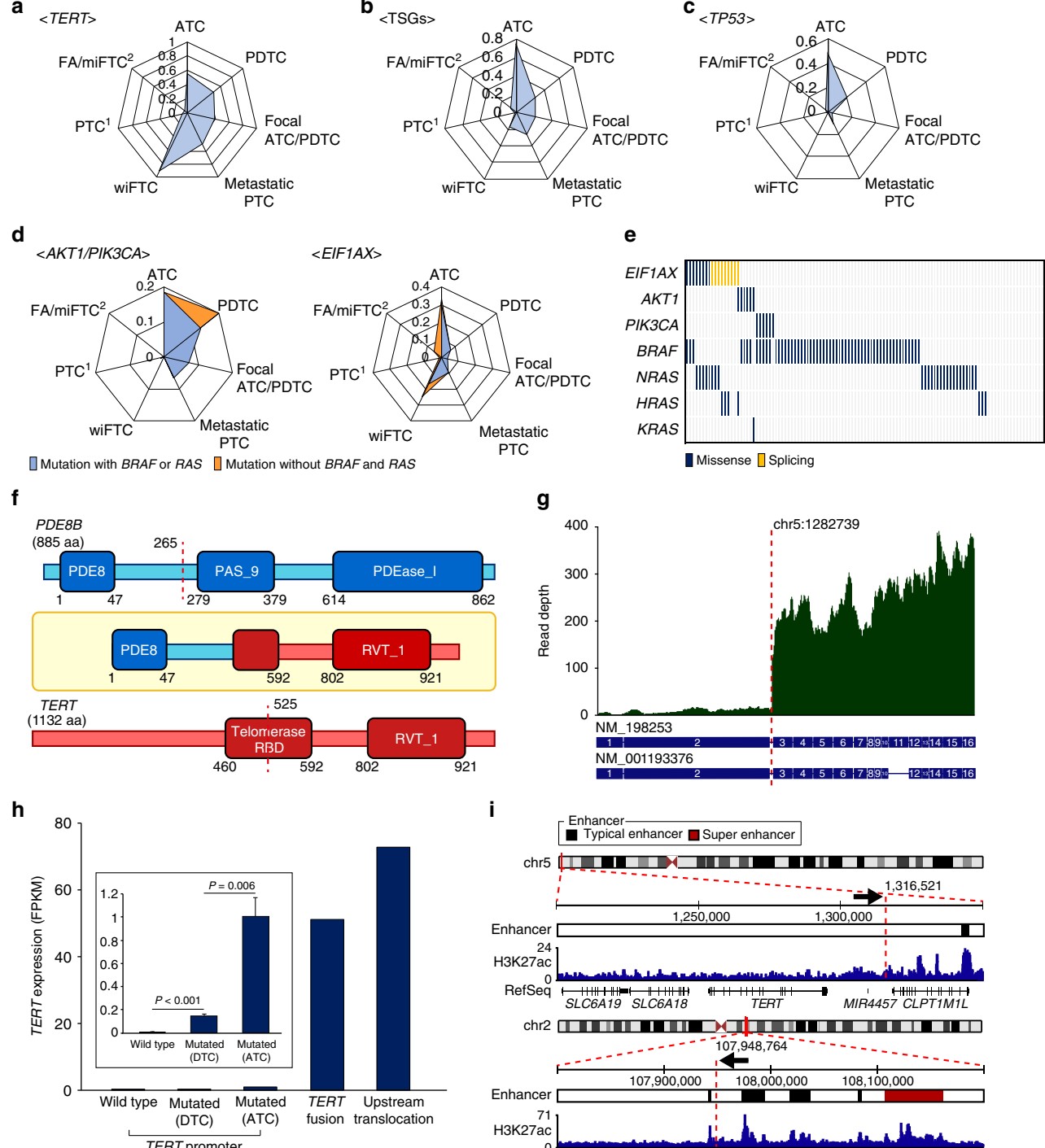

**Fig. 2** Recurrently altered genes in ATC and advanced DTCs. The frequencies of recurrent gene alterations among diverse types of TC were represented by radar charts. **a** *TERT*, **b** TSGs, **c** *TP53*, and **d** the frequencies of *AKT1/PIK3CA* and *EIF1AX* co-mutations with *BRAF*[V600E] or *RAS* in diverse types of TC (blue). The mutational frequencies without *BRAF*[V600E] and *RAS* were also displayed (orange). **e** The distribution of recurrent oncogene mutations across 113 TCs. **f** The functional domains of *PDE8B*–*TERT* fusion. **g** The expression level of *TERT* at each exon with *PDE8B-TERT* fusion. **h** The expression level of *TERT* by alteration type. *P*-values from DESeq2 were represented. **i** A schematic illustration of *TERT* upstream translocation. The breakpoints were pointed out by arrows

**Other genomic characteristics of ATC and advanced DTCs.** Tumor mutational burden (TMB) of ATC was higher than that of other types of TC ($P < 0.001$ for each; Supplementary Fig. 8). However, wiFTC did not show a higher TMB than FA/miFTC ($P = 0.40$). Using whole-genome sequenced tumors, we assessed the mutational signature, which suggests the particular mutagenesis processes in the cancer genome[20]. We found that most tumors (15/16) had signature 5, the most common signature in many cancers with unknown etiology (Supplementary Figs. 9 and 10). Meanwhile, one focal ATC/PDTC presented signature 2, which is associated with the activation of the AID/APOBEC family of cytidine deaminases[20].

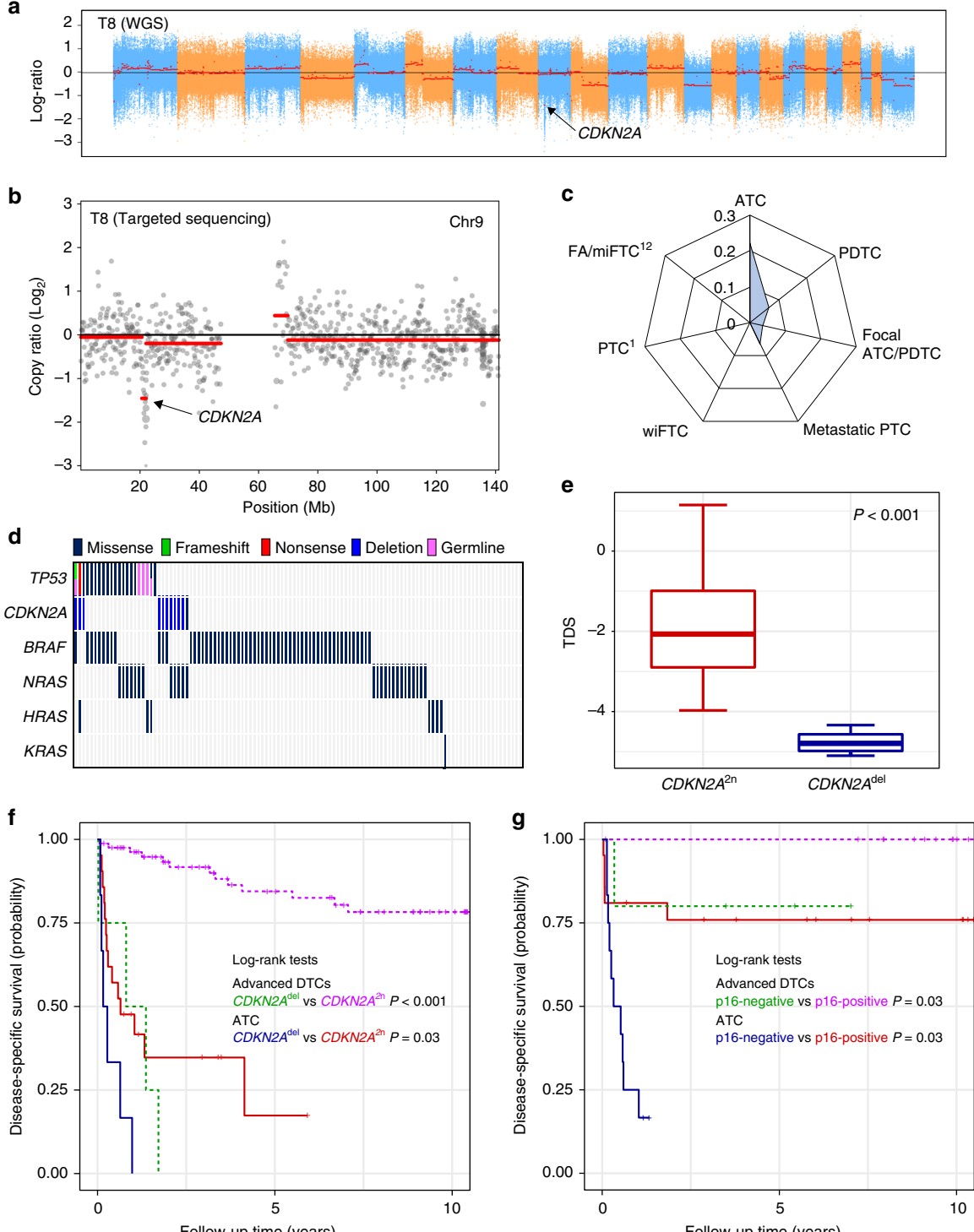

**Fig. 3** *CDKN2A* loss in ATC and advanced DTCs. **a** Detection of *CDKN2A* loss in ATC using WGS. **b** Detection of *CDKN2A* loss in ATC using targeted sequencing. **c** The frequency of *CDKN2A* loss among diverse types of TC were represented by a radar chart. **d** The distribution of *TP53* and *CDKN2A* alterations across 113 advanced TCs. **e** The effect of *CDKN2A* loss on thyroid differentiation score (TDS) in ATC. *P*-values from two-tailed Mann–Whitney *U*-test were represented. **f** The effect of *CDKN2A* loss on disease-specific survival in patients with ATC and advanced DTCs. del and 2n indicate *CDKN2A* copy number loss and neutral, respectively. **g** The effect of p16 expression on disease-specific survival in patients with ATC and advanced DTCs. *P*-values from Log-rank test were represented

The arm-level somatic copy number alterations (SCNA) burden was higher in ATC relative to DTCs (Supplementary Fig. 11a). In particular, $BRAF^{V600E}$-positive ATC showed a dramatic increment of arm-level SCNA compared with $BRAF^{V600E}$-positive PTC[21] ($P < 0.001$; Supplementary Fig. 11b). $RAS$-positive ATC also had a higher burden of arm-level SCNA than $FA/miFTC^{12}$ ($P = 0.01$; Supplementary Fig. 11c). wiFTC showed a similar level of arm-level SCNA as ATC ($P = 0.59$).

**Prognostic significance of *CDKN2A* loss for disease-specific mortality**. Furthermore, we discovered the recurrent copy number altered regions in ATC (Supplementary Table 1). Among them, 9p21.3 was the only region that covers cancer-related genes, *CDKN2A* and *CDKN2B* (Fig. 3a). It was the most significant copy number loss region, and we also successfully identified *CDKN2A* loss using targeted sequencing approach (Fig. 3b). *CDKN2A* was the second most frequently altered TSG in ATC, with a frequency of 22.22% (Fig. 3c), consistent with a previous study[9] (23.47%; Supplementary Fig. 1). Most *CDKN2A* alterations (72.73%) occurred without *TP53* mutations (Fig. 3d). *CDKN2A* loss was also identified in advanced DTCs (6.67%, 3.56%, and 6.45% in PDTCs, focal ATC/PDTCs, and metastatic PTCs, respectively), but not in FA/miFTC and PTC[1,12]. We also found one missense mutation (T79I) in focal ATC/PDTC.

Intriguingly, thyroid differentiation score (TDS) in ATCs with *CDKN2A* loss was significantly lower than those with wild-type ($P < 0.001$; Fig. 3e). We also found that *CDKN2A* loss was significantly associated with increased disease-specific mortality in patients with ATC and advanced DTCs ($P = 0.03$ and $P < 0.001$ for each; Fig. 3f). The hazard ratios (HRs) were 6.67 (95% confidence interval [CI], 1.34–33.12) and 9.88 (95% CI, 1.97–49.57), respectively, after adjustments for the age at surgery, sex, distant metastasis, and tumor origin (Supplementary Table 2). However, other recurrently mutated genes were not closely associated with patient survival (Supplementary Figs. 12 and 13).

We validated the relationship between *CDKN2A* loss and patient outcome by p16 immunohistochemistry (IHC) using tissue microarray (TMA) of 57 ATC and advanced DTC samples, of which 42 were sequenced samples (Supplementary Fig. 14a). All tissues (5/5) with *CDKN2A* loss, and 43.2% of *CDKN2A* wild-type (16/37) were p16-negative (Supplementary Fig. 14b). p16 expression displayed the association between poor disease-specific survival in patients with ATC and advanced DTCs ($P = 0.03$ for both; Fig. 3g). In ATC, p16-negative status increased the risk of disease-specific mortality (HR, 35.25; 95% CI, 1.38–898.79) after adjusting the age at surgery and sex, although the statistical significance was lost without adjustment for covariates or after additional adjustments for distant metastasis and tumor origin (Supplementary Table 3). In advanced DTCs, the hazard ratio was not calculated since p16-positive patients were all censored.

**Transcriptome landscape of ATC and advanced DTCs**. Transcriptome of 13 ATCs, 3 focal ATC/PDTCs, and 9 wiFTCs which were sequenced by WGS or WES were profiled by RNA-seq, then compared with data from 162 DTCs of our previous study[2]. Using *K*-means clustering via principal component analysis (PCA), the molecular subtype of each tumor was determined. The majority of the ATCs (10 of 13) showed a clear separation from the *BRAF*^V600E-like, *RAS*-like, and NBNR which are three molecular subtypes of DTC, and formed a fourth cluster (Fig. 4a). We designated this molecular subtype as ATC-like, and this cluster did not show the molecular distinctions resulting from the types of driver mutations that have been demonstrated in DTC[1,2]. One ATC with no alteration in cancer-related genes was also classified as ATC-like, and showed similar DNA methylation patterns to those of other ATC-like tumors (Supplementary Fig. 15). The expression profiles of focal ATC/PDTCs seemed to be dominated by the preceding DTC, since they were all classified as *BRAF*^V600E-like.

TCGA's *BRAF*^V600E-*RAS* score (BRS) analysis displayed a similar result to that of *K*-means clustering via PCA; *BRAF*^V600E-positive and *RAS*-positive ATCs displayed no variance in BRS ($P = 0.91$; Fig. 4b), in contrast to PTC ($P < 0.001$). For wiFTC, one tumor was classified as ATC-like by *K*-means clustering via

PCA, but most of them were *RAS*-like or NBNR (Fig. 4a). However, BRS analysis suggested that wiFTC is differentiated from FA/miFTC on the molecular level ($P = 0.001$; Fig. 4b). Then, we performed TDS and ERK score analyses, which illustrate thyroid differentiation and the activity of the MAPK-signaling pathway, respectively[1]. Lower TDS and higher ERK scores were observed in ATC than in DTC (Fig. 4c). ATC showed similar thyroid differentiation regardless of the driver mutation ($P = 0.67$; Fig. 4d), and MAPK-signaling pathway was even more activated in *RAS*-positive ATC than in *BRAF*^V600E-positive ATC ($P = 0.02$; Fig. 4e). Unlike ATC, ERK score was not higher in wiFTC ($P = 0.88$), but TDS was significantly lower ($P < 0.001$) compared with FA/miFTC. In wiFTC, the expression of seven genes (*SLC5A8*, *TPO*, *FOXE1*, *DIO2*, *TG*, *GLIS3*, and *SLC26A4*) that are related to thyroid metabolism and function was significantly reduced (Fig. 4f). Furthermore, most of thyroid metabolism genes were repressed in ATC, except for three genes (*THRA*, *NKX2-1*, and *PAX8*).

**Potential druggable targets in ATC**. Kyoto Encyclopedia of Genes and Genomes (KEGG) pathway[22] enrichment analysis using differentially expressed genes (DEGs) was performed to discover the biologically relevant pathways that are regulated during the progression of TC. The alterations of specific pathways during the progression of *BRAF*^V600E-positive or *RAS*-positive ATC were demonstrated separately. We found that various pathways, such as the MAPK-signaling pathway, focal adhesion, extracellular matrix (ECM) receptor interaction, p53 signaling, and cell adhesion molecules (CAMs), which were initially increased in PTC relative to the normal thyroid, were further activated in *BRAF*^V600E-positive ATC (Fig. 5a). Notably, the VEGF-signaling pathway and the Notch-signaling pathway, which are closely associated with angiogenesis[23,24], but were not activated in PTC, were significantly elevated in *BRAF*^V600E-positive ATC.

In *RAS*-positive ATC, the MAPK-signaling pathway, focal adhesion, ECM receptor interaction, p53-signaling pathway, cell cycle, and CAMs were also further elevated compared with three types of follicular-patterned thyroid tumors (Fig. 5a). Except for the MAPK-signaling pathway, these pathways were not activated in wiFTC relative to FA/miFTC. Moreover, the JAK-STAT-signaling pathway, which was not detected in the three types of follicular-patterned tumors, was activated in *RAS*-positive ATC (Fig. 5b). In order to evaluate the potential ability of activated pathways in ATC as druggable targets, we performed functional in vitro experiments with ATC cell lines. In *BRAF*^V600E-positive ATC cell lines, we were unable to demonstrate effects of inhibition of VEGF-Notch signaling. Whereas, in CAL62, the *RAS*-positive ATC cells, JAK inhibition with ruxolitinib decreased the expression of *SOCS3*, *BCL2L1*, and *MYC* which are the downstream molecules of JAK-STAT pathway (Fig. 5c), and then we confirmed reduced cellular proliferation (Fig. 5d).

We also found that the calcium-signaling pathway and various pathways related to metabolism, such as glycerolipid metabolism and fatty acid metabolism, were down-regulated in ATC compared with DTC (Supplementary Fig. 16). In addition to the aforementioned pathways, the expression levels of *CD274* and *PDCD1LG2*, which encode PD-L1 and PD-L2, respectively, were increased in some ATCs (Fig. 5e). Interestingly, the up-regulation of these genes were found in ATCs with *CDKN2A* loss (Log$_2$fold-change [FC] = 2.38 and 2.85 for each; Fig. 5f).

**Discussion**

In this study, we elucidated the genomic and transcriptomic landscape of ATC and advanced DTCs using diverse types

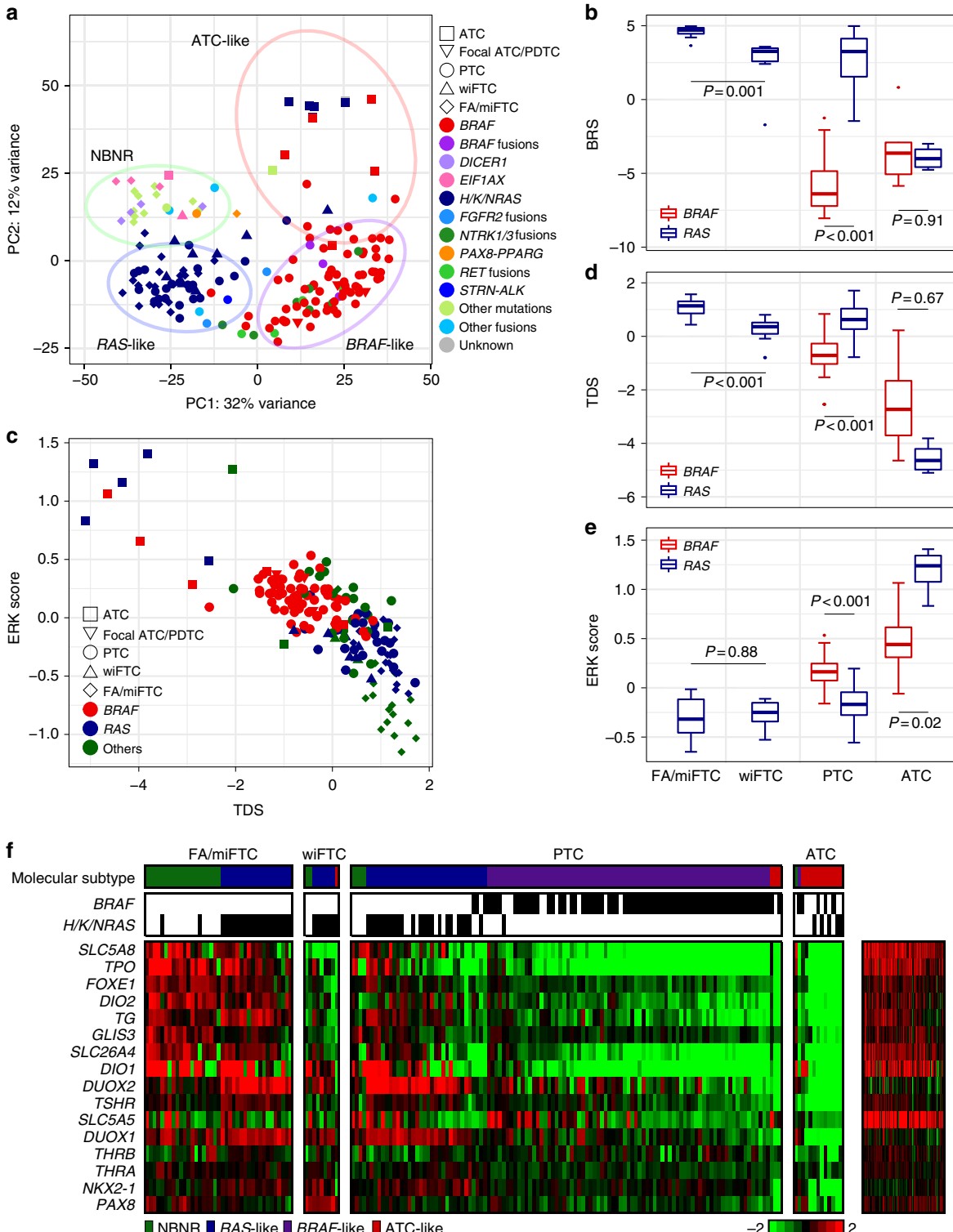

**Fig. 4** The transcriptome landscape of ATC and advanced DTCs. **a** The result of *K*-means clustering via PCA. The types of tumors and driver mutations were represented by shape and color, respectively. **b** The results of *BRAF*$^{V600E}$-RAS score (BRS) analysis were represented by box plots. *P*-values from two-tailed Mann–Whitney *U*-test were represented. **c** TDS and ERK score were displayed on a scatter plot. **d** The results of TDS analysis and **e** ERK score analysis were represented by box plots. *P*-values from two-tailed Mann–Whitney *U*-test were represented. **f** The heatmap represents the expression profile of 16 genes associated with thyroid function and metabolism in TCs. The right panel represents gene expression levels in normal thyroid tissues

of sequencing technologies. Our mutational profiling confirmed that multiple hits of genetic alterations promote the progression of TC, as described in previous reports[8,9]. Moreover, we were able to extend the current state of

knowledge about the transcriptomic characteristics of advanced TCs.

Several studies demonstrated the association between *TERT* promoter mutation and aggressive clinicopathological features of

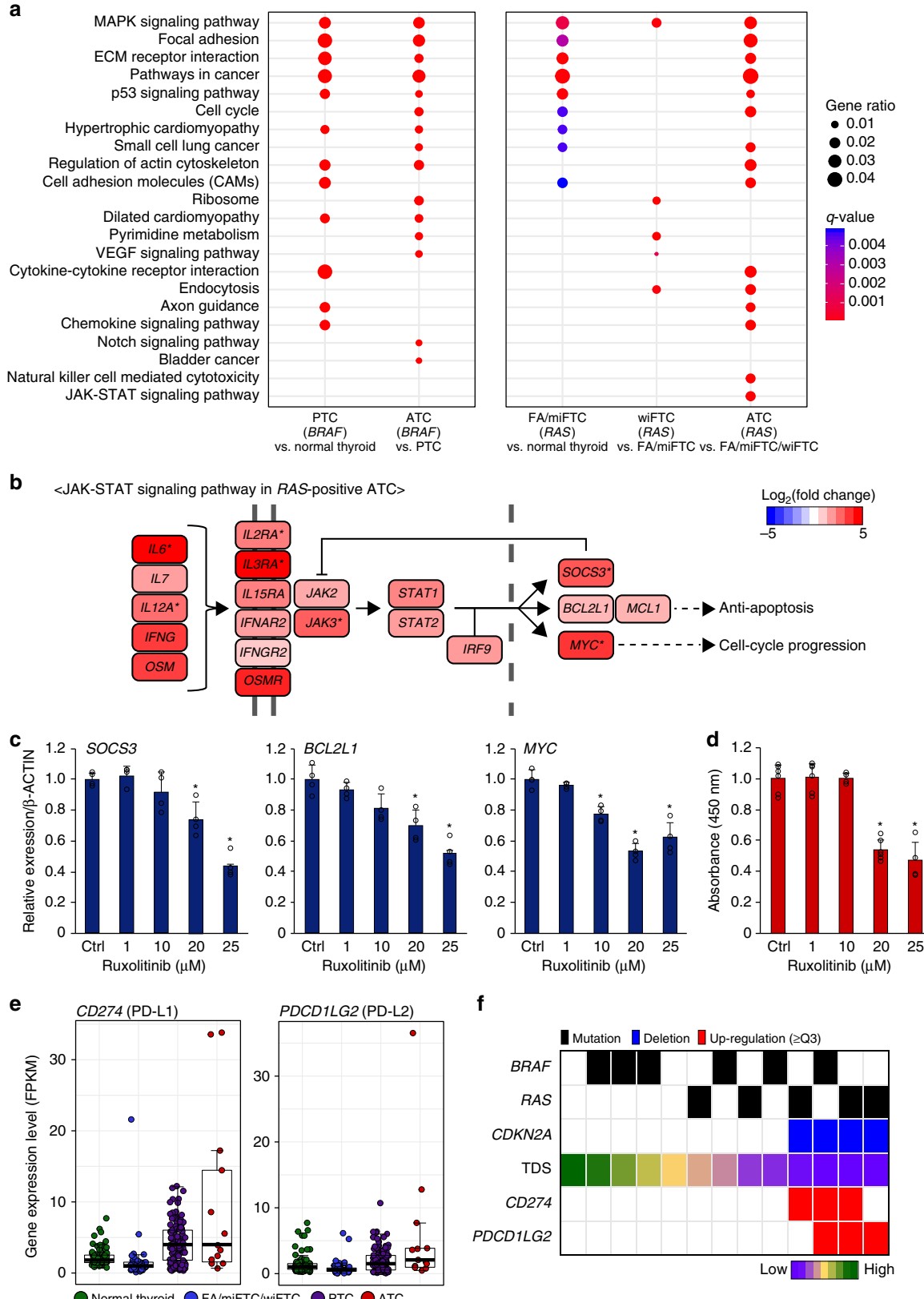

TC[25–27]. In this study, *TERT* was the most frequently co-mutated gene in both ATC and advanced DTCs in addition to main driver genes (*BRAF*[V600E] and *RAS*). Especially, 91.67% of wiFTCs harbored *TERT* alterations, including promoter mutations and two rearrangements, but only 25.00% of them harbored the additional mutation in an oncogene or TSG. Therefore, this result again underscores the importance of *TERT* in metastatic, invasive, and early aggressive characteristics of DTC, rather than anaplastic transformation of TC. The activation of *TERT* in cancer was thought to be caused by point mutations in its promoter region[28].

**Fig. 5** The potential druggable targets of ATC. **a** The top 15 significantly up-regulated KEGG pathways in *BRAF*[V600E]-positive and *RAS*-positive ATCs. The significance of these pathways were also noted in PTC, FA/miFTC, and wiFTC, as they were also found within the top 15 significantly up-regulated pathways of each tumor. **b** The increment levels of genes in the JAK-STAT-signaling pathway in *RAS*-positive ATC compared with FA/miFTC/wiFTC were represented by $\log_2$(fold-change) values. Asterisks indicate the genes that were also up-regulated *BRAF*[V600E]-positive ATC. **c** Quantitative reverse transcription polymerase chain reaction measurement of expression of JAK-STAT-signaling pathway genes and **d** cell viabilities analyzed by cell counting kit-8 assay, in CAL72 cells following treatment with ruxolitinib (1, 10, 20, and 25 μM). Ctrl denotes control. All data were expressed as mean ± standard deviation. *$P < 0.05$ from two-tailed Mann–Whitney $U$-test (compared with controls). **e** The expression levels of *CD274* and *PDCD1LG2* in various types of TC. **f** The relationship between the up-regulation of two immunotherapeutic genes and *CDKN2A* loss. Samples were sorted by high to low TDS

However, structural rearrangements were also reported to trigger the extreme up-regulation of *TERT* and be associated with tumor aggressiveness[18,19,29]. Likewise, we determined the high expression level of *TERT* in two tumors with *PDE8B-TERT* fusion gene and inter-chromosomal translocation, t(2;5)(2q;5p), in *TERT* upstream region. Although two *TERT* rearrangements across 113 TCs were found, there might be more tumors with these alterations, since targeted sequencing method which was performed on most of study subjects (77.88%) did not cover intronic and intergenic regions.

We also confirmed that the prevalence of second-hit and third-hit in oncogenes and TSGs were increased in ATC. Intriguingly, co-mutations in the oncogenes, *AKT1/PIK3CA* or *EIF1AX*, were frequently discovered in ATC and advanced DTCs with *BRAF*[V600E] or *RAS* mutations, respectively. This suggests that the advancement of TC can be predicted by different markers according to the main driver mutation of DTC. *TP53* and *CDKN2A* were the most frequently altered TSGs in ATC and advanced DTCs, as described in a recent report[9]. The prognostic significance of *CDKN2A* has been well described for diverse types of cancer[30,31], but not yet for TC. Using TDS analysis, we found that ATC with *CDKN2A* loss presented the poorest thyroid differentiation. Moreover, we demonstrated an association between *CDKN2A* loss and increased disease-specific mortality in patients with ATC or advanced DTCs, even after adjustment for the other potential prognostic factors.

Altogether, our results exhibited the potential contribution of *TERT* and diverse oncogenes (*AKT1/PIK3CA* and *EIF1AX*) in the early progression of DTC, and higher relation of TSGs (e.g., *TP53* and *CDKN2A*) in anaplastic change of TC. Notably, *CDKN2A* loss may be a strong prognostic factor for patients with advanced DTCs as well as ATC.

It is well established that the molecular characteristics of DTC are determined by the types of driver mutations[1,2]. Two molecular subtypes, *BRAF*[V600E]-like and *RAS*-like, were first proposed by TCGA[1], and our group showed the existence of a third subtype, NBNR[2]. Based on molecular subtype, TC displays differential regulation of signaling pathways. Landa et al. reported that *RAS*-positive ATCs lost the molecular characteristics that were exhibited in *RAS*-positive DTC and that all ATCs characterized to be *BRAF*[V600E]-like, regardless of the driver mutation[8]. The remarkable transcriptomic changes in *RAS*-positive ATC were also confirmed in this study. Moreover, we were able to demonstrate the extended molecular perspective of ATC with the support of our previously published RNA-seq data from 162 DTCs[2]; both *BRAF*-positive and *RAS*-positive ATCs presented a similar transcriptome profile, irrespective of their driver mutation, but were grouped into another subtype, ATC-like, rather than *BRAF*[V600E]-like.

In addition to ATC-like, we identified potential druggable pathways in ATC. In both *BRAF*[V600E]-positive and *RAS*-positive ATCs, the MAPK-signaling pathway and several cell–cell communication pathways were further up-regulated compared with DTC. Furthermore, we found the activation of VEGF and Notch-signaling pathways in *BRAF*[V600E]-positive ATC and the

JAK-STAT-signaling pathway in *RAS*-positive ATC, which were not activated in DTC. Although we demonstrated that the cell viability was regulated by inhibition of the activated JAK-STAT signaling in *RAS*-positive ATC cell line in vitro, inhibition of Notch signaling did not affect *BRAF*[V600E]-positive ATC cell lines. It is possible that the effect might not have been shown, since there would be other pathways that regulate VEGF-Notch signaling[32]. In addition to the aforementioned pathways, previous reports showed that cell lines with *CDKN2A*/p16 loss is linked to response to CDK4/6 inhibitors, such as palbociclib[33–35] and to resistance to *BRAF*[V600E]-selective inhibitor, vemurafenib, in metastatic *BRAF*[V600E] PTC cells[36]. Therefore, our findings would give clues to choose appropriate target agents for ATC.

A putative association between the APOBEC family of cytidine deaminases and the progression of ATC was recently reported, but it was demonstrated by an analysis based on only a small number of variants (8–20)[9]. In this study, we clearly showed the presence of mutational signature 2 in one focal ATC/PDTC with over 10,000 variants that were discovered by WGS. Hence, targeting APOBEC mutagenesis may be an option for treatment of a few advanced TCs.

Immunotherapy is the most promising state-of-the-art for cancer therapy[37]. Growing evidence indicates that patients who have a tumor with a high TMB are highly responsive to immunotherapy[38,39]. We also found a higher TBM in ATC than in DTC, consistent with recent studies[8,9]. Furthermore, ATCs with *CDKN2A* loss displayed up-regulation of *CD274* (PD-L1) and *PDCD1LG2* (PD-L2), which are favorable immunotherapeutic targets for treating cancer[37], in agreement with a previous report that showed the expression of PD-L1 in ATC[40]. The underlying reason of the increased expression of *CD274* and *PDCD1LG2* in ATC with *CDKN2A* loss is not unveiled in this study, but their potential relationship in non-small cell lung cancer were previously reported[41,42]. Put together, immunotherapy for ATC might be effective, considering the increased TMB and expression of PD-L1/PD-L2, especially for ATC with *CDKN2A* loss.

In conclusion, this study presented a comprehensive analysis of the genomic and transcriptomic alterations associated with the progression of DTC to its advanced and anaplastic types. We expect that our findings will provide more tailored diagnostic and therapeutic interventions for these fatal diseases.

## Methods

**Ethics statement.** This study was approved by the institutional review board of Seoul National University Hospital, in accordance with the Declaration of Helsinki (approved ID: H-1307-034-501). Written informed consents were obtained from all patients.

**Patients.** Fresh frozen or formalin-fixed paraffin-embedded (FFPE) tissues from 113 patients with advanced TCs, including 27 ATCs, 15 PDTCs, 28 focal ATC/PDTCs, 12 wiFTCs, and 31 metastatic PTCs, who underwent thyroidectomy were analyzed using massively parallel sequencing method. The clinicopathological characteristics of 113 patients according to histology are shown in Table 1 and the clinical information of individual patient is provided in Supplementary Data 2. Median follow-up duration was 38.0 (interquartile range, 11.0–109.2) months and mean age of patients was $60.7 \pm 14.1$ years. Patients with ATC showed the highest

mortality rate (74.1%) and the shortest disease-specific survival (median, 6.9 months), while those with PDTC and focal ATC/PDTC showed higher rates of no evidence of disease (73.3% and 67.9%, respectively) and relatively longer disease-specific survival (60.3 and 109.2 months, respectively) compared to other histologic types of advanced thyroid cancer. For transcriptomic profiling, RNA-seq data of 162 patients with DTC from our previous study[2] were applied as reference dataset for newly sequenced 25 samples.

**Pathological diagnosis.** For the purposes of accurate diagnosis, previous pathologic specimens were re-evaluated by an experienced pathologist (J.-K.W.). ATC or PDTC was defined as a tumor in which ≥10% of its volume was occupied by undifferentiated or poorly differentiated cells, while the focal ATC/PDTC was defined as a tumor in which <10% of the tumor volume was occupied by undifferentiated or poorly differentiated cells in the background of differentiated cancer[43,44]. As there is no definitive established pathologic definition for this, the definitions adopted in this study were based on the experience of clinicians and pathologists. The ATC component in tumors mixed with DTC was defined based on the following features: the nuclei without the characteristic features of DTC and showing a greater ratio of nucleus/cytoplasm, nuclear pleomorphism other than the features of DTC, and a more solid growth pattern with or without p53 expression. PDTC was defined on the basis of the Turin proposal for the use of uniform diagnostic criteria[45], and was confirmed if showing a solid, trabecular, or insular growth pattern with the absence of conventional nuclear features of papillary carcinoma, and the presence of at least one of the following features: tumor necrosis, mitotic count ≥3/10 high-power field, or convoluted nuclei. wiFTC was defined in the case of widespread infiltration of thyroid tissue and/or vascular invasion according to WHO criteria[46]. All metastatic PTC accompanied distant metastasis in other organs and 77.42% of them (24/31) also had lymph node metastasis.

**Massively parallel DNA sequencing.** We performed WGS using Hiseq X instrument (Illumina, San Diego, CA, USA). For WGS, fragmentation of gDNA samples (except for T75 and N75) followed standard Illumina protocols except for an additional restriction enzyme digest step at the beginning of the work flow: a 1 µg of the DNA was first cut with a single methyl-sensitive restriction endonuclease, HpaII. Once digested, DNA was washed with Qiagen's QIAquick PCR Purification Kit and sheared to a median size of 300 bp using a Covaris S220. Tumor and matched normal samples (except for T1) were sequenced with average sequencing depth of 72.23X and 35.74X, respectively.

WES (including WGS of T75 and N75) and targeted sequencing were performed with Hiseq 2000 and 2500 instruments, respectively (Illumina, San Diego, CA, USA), according to manufacturer's instruction. For WES, we captured whole-exome region using SureSelect Human All Exon V4 kit, and tumor and matched normal samples were sequenced with average depth of 129.27X and 138.54X, respectively. Also, we downloaded the previously published WES data regarding PTC (n = 28) and FA/FTC (n = 18) for comparative analyses[12,21]. Paired-end reads were aligned to GRCh37.p13 reference using BWA[47]. Duplicated reads were removed by Picard tools (see URLs). Insertion/deletion (indel) realignment and base quality score recalibration (BQSR) was performed by Genome Analysis Tool Kit (GATK)[48].

For targeted sequencing, we designed custom DNA capture probes using the Agilent SureDesign (see URLs). This custom probes contained the exonic regions of 57 genes, 13 regions for fusion gene rearrangements, and one region for TERT promoter mutation (Supplementary Table 4). The target genes were selected by our preliminarily findings from WGS/WES and the previously reported genes in TC[1,2,8,49,50]. Moreover, four genes (STARD9, HUWE1, BAZ2B, and MCM6) which were discovered in our unpublished work about distant metastasis of FTC were included. The average sequencing depth of 421.54X were achieved and matched normal samples were not included.

**Massively parallel RNA sequencing.** We conducted RNA-seq on 13 ATCs, 3 focal ATC/PDTCs, and 9 wiFTCs which were also sequenced by WGS and WES, respectively. Read alignment and gene expression quantification were performed using STAR[51] and HTSeq[52], respectively. DEGs were found with DESeq2[53] as following criteria: (1) adjusted P < 0.05 and (2) |Log₂FC| ≥ 1. For DEG analysis, we only used BRAF^V600E-positive classical PTC and RAS-positive FA/miFTC as comparison groups for BRAF^V600E-positive ATC (PTC origin) and RAS-positive ATC (FTC origin), respectively. Then, DEGs were subjected to KEGG pathway enrichment analysis by Molecular Signatures Database[22,54]. As massively parallel DNA sequencing, we also performed indel realignment and BQSR using GATK[48].

**Scoring analysis.** We used three scoring analysis methods from TCGA study[1]: TDS, ERK score, and BRS. For TDS, 16 genes that are related to thyroid function and metabolism were subtracted by median across all tumors samples and their average value were used. For ERK score, 52 MAPK-signaling pathway genes subtracted by median across all tumors and their average value were used. For BRS, single sample gene set enrichment analysis[55,56] from GenePattern (see URLs) was performed with 71 gene signatures from TCGA study.

**Variant detection.** For all sequencing methods, MuTect and GATK's HaplotypeCaller were applied to discover single nucleotide variant (SNV) and indel, respectively[48,57]. For MuTect, we applied somatic detection mode when matched normal sample is available. All variants were annotated by ANNOVAR[58]. To determine variants that are putatively associated with the progression of TC, we only kept variants which are not commonly found in The Exome Aggregation Consortium database (minor allele frequency <0.1%)[59]. Furthermore, missense SNVs which were functionally predicted as deleterious by PolyPhen2 or SIFT[60,61], and the loss of function variants, such as nonsense, splicing site, and frameshift variants were retained for subsequent analysis. Then, we separated genes into two groups to avoid potential bias derived from the absence/presence of matched normal samples as follows: (1) genes that were confirmed to have somatic mutations from WGS/WES analysis (Fig. 1) and (2) genes that were highly suspected to have germline mutations (Supplementary Fig. 1).

**Structural variation detection.** To discover SCNA, we used FACETS and EXCAVATOR2 for WGS and WES data, respectively[62,63]. Then, GISTIC2 deduced the arm-level SCNA and the significantly altered chromosome regions[64]. For targeted sequencing data, CNVKit was implemented excluding antitarget regions with parameter '-m haar' in segmentation step to determine CDKN2A deletion[65]. We considered the segment with Log₂ratio ≤−0.6 as homozygous deletion, since this threshold reproduced the consistent results compared with the results from WGS. For targeted sequencing data, we discovered fusion genes when more than five split or discordant reads were found. For RNA-seq data, we used MOJO (see URLs) to identify fusion genes.

**Mutational signature analysis.** To assign specific mutational signature of each cancer genome, the average hierarchical clustering analysis was performed using mutational signatures from whole-genome-sequenced samples with 30 reference signatures. The mutation type probabilities of 96 motifs were extracted by SomaticSignatures[66]. The reference signatures were downloaded from COSMIC (see URLs).

**Enhancer prediction.** We used H3K27ac chromatin immunoprecipitation sequencing (ChIP-seq) data to identify enhancers in human thyroid gland. The ChIP-seq data of thyroid gland from 54 years old male were downloaded from ENCODE project[67]. We defined super-enhancers as described in other study[68]. In short, 76 bp single-end reads were aligned to the GRCh37 reference genome using bowtie 0.1.1 using following parameters: '-k 2', '-m 2', '-n 2', '-S', and '−best'[69]. Then, MACS 1.4.2 identified the enrichment regions of H3K27ac in thyroid gland following parameters: '-p 1e-9', '−keep-dup = auto', '-w −S −space = 50', and '-g hs'[70]. Using output of MACS, ROSE identified the super-enhancers[71]. The constituent enhancers were stitched together if they are within 12,500 bp. If constituent enhancers were fully contained within promoter region (window ± 2000 bp from the transcription start site), they were excluded from stitching. At last, we separated super-enhancers and typical-enhancers from each other by isolating an inflection point of H3K27ac signal versus enhancer rank.

**DNA methylation analysis.** HpaII, methylation-sensitive restriction enzyme, recognizes and digests CCGG sites, if second cytosine is unmethylated[72]. Therefore, global DNA methylation pattern could be investigated using WGS with assistance of this characteristic of HpaII. After alignment of HpaII-digested paired-end reads to GRCh37 reference genome, we extracted the mapped reads which span CCGG sites of GRCh37. Then, we calculated DNA methylation level of second cytosine based on 10 possible patterns of spanning read. For this, only reads with high mapping quality (≥20) were used. Four patterns which span CCGG site but do not contain information of DNA methylation status were excluded from calculation. At last, we analyzed only CCGG sites that were appropriately covered by mapped reds (≥5) across 16 ATC tumors.

**Polymerase chain reaction analysis and Sanger sequencing.** TP53 germline mutations, TERT promoter mutations, and TERT rearrangements were confirmed by PCR and Sanger sequencing. The PCR was performed by using a StepOne Plus real-time PCR system (Applied Biosystems, Foster City, CA, USA). The PCR primer sequences are listed in Supplementary Table 5. Sanger sequencing with PCR product was conducted with a BigDyeTM Terminator Cycle Sequencing Kit (Applied Biosystems, Foster City, CA, USA) using an ABI 3730XL Genetic Analyzer (Applied Biosystems, Foster City, CA, USA). For TERT promoter mutations, Primer #1 was mainly used and when the result was not clear, we used Primer #2 to confirm the result. It was performed on all the subject in this study, as well as FAs and miFTCs which were included in our previous study[2].

**TMAs and immunohistochemical staining.** For the IHC analysis of p16, we constructed TMAs including 17 ATCs and 40 advanced DTCs (15 PDTCs and 25 focal PD/ATCs). TMAs were constructed from 2-mm-diameter cores derived from representative tumor areas of FFPE tissue blocks. p16 IHC was performed using the DAKO Omnis autostainer (DAKO-Agilent Technologies, Santa Clara, CA, USA)

with a mouse anti-p16 monoclonal antibody (1:4, F. Hoffmann-La Roche, Basel, Switzerland).

**Cell cultures and in vitro inhibition assays**. CAL62, a human ATC cell line harboring $KRAS^{G12R}$ were kindly provided by Dr. Yong-Hyun Jeon (Kyungpook National University Hospital, Daegu, Republic of Korea). CAL62 cells were cultured in Dulbecco's Modified Eagle Medium supplemented with 10% of fetal bovine serum and incubated at 37 °C in a humidified atmosphere containing 5% of $CO_2$. Approximately $0.8 \times 10^5$ cells/mL were then seeded on a 12-well culture plate. After 12 h, CAL62 cells were treated with ruxolitinib (1, 10, 20, and 25 μM; Selleckchem, Houston, TX, USA), a JAK1/2 inhibitor. After 24 h of treatment, mRNAs were extracted for the analysis of expression levels of downstream genes of JAK-STAT- signaling pathway. The primers are listed in Supplementary Table 5. After 48 h of treatment, cell viability was then determined by using a CCK-8 assay (Dojindo, Kumamoto, Japan). The CCK-8 solution (50 μL) was added to each well and, after 50 min incubation, absorbance at 450 nm was measured with a microplate reader (Molecular devices, San Jose, CA, USA).

**Statistical analyses**. For statistical comparisons, two-tailed Mann–Whitney $U$-test was used by SPSS 23.0 (IBM Co., Armonk, NY, USA). Disease-specific survival curves were generated by Kaplan–Meier method and compared by Log-rank test using survfit() function of survminer in the R programming language. For Cox proportional hazard model, coxph() function was applied.

**URLs**. TCGA Fusion Gene Database. Agilent SureDesign. Picard tools, GenePattern, MOJO, COSMIC.

**Reporting summary**. Further information on research design is available in the Nature Research Reporting Summary linked to this article.

## Data availability

All sequencing reads were submitted the European Genome Phenome Archive (https://www.ebi.ac.uk/ega/) with accession number EGAS00001003540.

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

## Acknowledgements

This research was supported by the Basic Science Research Program through the National Research Foundation of Korea, funded by the Ministry of Science, ICT & Future Planning (grant number: NRF-2016R1A2B4012417).

## Author contributions

J.-S.S. and Y.J.P. conceived and designed study. S.-K.Y. and Y.S.S. wrote the manuscript. S.-K.Y. and J.H. performed bioinformatic analyses. H.H.K. and G.J. performed *TERT* promoter sequencing and PCR. J.-K.W. contributed to the histopathological review. J.-Y.S. performed massively parallel sequencing experiments. E.K.L., Y.A.K., S.-j.K., S.W.C., E.-J.C., K.E.L. and J.-I.K. contributed to data interpretation.

## Additional information

**Competing interests:** The authors declare no competing interests.

