## [Peer Review File · Nature Communications]

Reviewers' comments:

Reviewer #1 (Remarks to the Author):

In the present manuscript the Authors investigated both mutational landscape and transcriptome of anaplastic thyroid cancer (ATC) and advanced differentiated thyroid cancer (DTC) (total 113 patients). By doing this they identified a novel ATC-like subtype of thyroid cancer. Moreover, they observed that CDKN2A losses to be associated with poor survival and upregulation of genes coding for PD-L1 and PD-L2. Finally, the Authors reports the activation of potentially targetable VEGF-notch and JAK-STAT signalling pathways in ATC.

The topic is certainly of interest, even if the DNA sequencing data are not novel as also stated by the Authors. Some of the results might be relevant to improve our knowledge of molecular mechanisms involved in thyroid cancer development and progression. There are also a number of interesting preliminary insights about potential future targeted therapies. The manuscript is generally well written and the bioinformatics analysis is well performed. However, I do have some major concerns, especially about the non-homogeneity of the sequencing methods used and the lacking of functional validation.

Major comments

General: some assertions in the Abstract and in the Discussion should be lowered down as there is no functional validation shown. For instance, from presented data, it is difficult to say "TERT induces early progression of DTC".

Methods:

-the tumour DNA-samples were investigated by using different methods: 17 by WGS, 8 by WES and 88 by targeted sequencing (panel of 57 genes, 13 regions for fusion rearrangements and one region for TERT promoter mutations), sometimes with different sequencing depth. The Authors should clearly state to what extent this variability might affect their results. Moreover, they should specify how did they chose the genes contained in the panel for targeted sequencing.

- it is not entirely clear how was selected the series for transcriptomic profiling and how the data were analysed. At line 344-345 "RNA-seq data of 162 patients with DTC from our previous study were analysed with those of advanced TCs". Do the Authors mean that the data from the previous 162 DTC were used as reference for the new series of 113 advanced TCs

- matched normal DNA was not analysed for the targeted sequenced tumors, which are actually the majority. This should be better specified in the Methods section. It can drive to some bias. For instance, as the Authors write in the results section, it is also possible that mutations in some genes such as KMT2D, ATM, CHEK2, NF1, NF2, MEN1 might be rare germline mutations. This may be true for TP53 gene as well. This might be a major issue and should be clearly stated and discussed.

- the relationship between presence of CDKN2A loss and short survival looks noteworthy from the survival curves (Fig 3g-h, n=27 ATC and n=86 advanced DTC). However, it is not enough to conclude that CDKN2A loss might represent a prognostic factor. A multivariate regression analysis including other potential prognostic factors (e.g. presence of distant metastasis at diagnosis? Histopathological parameters? Etc) should be added. The Hazard Ratio and the 95% confidence interval must also be included. Importantly, the effect of CDKN2A loss on survival should be validated on an independent cohort of ATC should be performed.

Results:

-did the Authors find any difference in terms of genomic alterations and/or transcriptome profile among samples coming from primary tumours, local recurrences or distant metastasis?

- lines 124-125: findings about mutations in ATRX, NFE2L2 and SPOP should be shortly presented in the text.

- CDKN2A losses (or mutations or promoter hypermethylation) are frequent in other types of tumours and have been deeply investigated. Indeed, it should be validated in ATC at the protein level by investigating p16 expression. CDKN2A losses also has been linked to response to CDK4/6

inhibitors (ex. Elvin et al 2017). This should be added.

Moreover, loss of CDKN2A has been previously linked to resistance to vemurafenib treatment in metastatic BRAF-PTC cells (Duquette et al 2015). This should be considered and at least discussed.

- line 241-242 and 243-244: from text and Fig 5d/5e, it is not fully clear in how many ATC samples were the expression of the genes coding for PD-L1 (CD274) and PD-L2 (PDCD1LG2) up-regulated. Moreover, I cannot see in Fig 5e the relationship between this overexpression and survival.

Finally, in the present study, there is no direct evidence to explain the connection between PD-L1 (CD274) and PD-L2 (PDCD1LG2) and CDKN2A losses.

-the Authors observed that genes involved in VEGF-notch and JAK-STAT signalling pathways are overexpressed in BRAF-positive ATC. In order to propose these potentially targetable pathways for developing future treatment options for ATC, specific functional experiments in ATC cell lines should be performed.

Minor comments:

Introduction, line 79: "to discover favourable targets for...." Should be modified into "to discover molecular mechanisms potentially involved in tumour progression and/or as targets for treatment".

Results, lines 88-89: the number of samples investigated by WGS, WES and targeted sequencing needs to be added here.

Results, line 180: how was exactly the "poorest thyroid differentiation" evaluated?

Legend to the Figures: TDS should be spelled.

Reviewer #2 (Remarks to the Author):

The study by Yoo et al describes a genomic analysis of several pathological types of follicular cell thyroid cancers, including the most lethal type, anaplastic thyroid carcinoma. The cohort for DNA sequencing consisted of primary tumors from thyroidectomies of 27 anaplastic cancers and 86 clinically advanced differentiated thyroid cancers that included 15 poorly differentiated carcinomas, 28 cases with focal anaplastic or poorly differentiated carcinoma, 12 widely invasive follicular carcinoma and 31 papillary carcinomas associated with metastatic disease. In addition, RNA sequencing data from a previous study was integrated in the analysis.

The resulting analysis is highly data rich and well presented. The information presented definitely advances the field, but there are some reasons for concern that need to be addressed.

The major criticism of the study relates to study design, specifically the variable use of WGS, WES and targeted sequencing across the different pathologic subtypes. For example, according to the legend in Figure 1, WGS was applied to ATC and a few "focal ATC/PDTC" and none of the PDTC or metastatic PTCs. WES was only applied to "WiFTCs" and, by default, targeted sequencing was only applied to PDTC and metastatic PTC. This design is clearly suboptimal as it introduces potential bias on the data because WGS will naturally find more variants than WES and targeted sequencing. In other words, they mixed the depth and range of sequencing across the different tumor histologies. This was likely not by design and should be explained how it came to be. It is not clear that this was normalized in the analysis, if possible. Was anything done to control or adjust for this? At least this should be discussed and highlighted as a potential limitation and source of bias.

Beyond the study design, I have several minor comments.

They use the BRAF-like and RAS-like distinction first developed by the TCGA study. They should use the more specific term, BRAF-V600E-like, because not all BRAF mutations have a BRAF-V600E-like phenotype.

The definition of focal ATC/PDCA is not clear and this should be defined using pathological terms.

The metastatic nature of the PTCs should be highlighted in the main text. Where these patients with distant metastatic disease or regional lymph nodes metastases. They should also make it very clear that they studied the primary tumors, not the metastatic tumors. It was confusing at first reading and I recognize there is data in Figure 1, but something more should be added in the Results to help the reader understand what was studied.

BRAF mutation is used in the paper when I think they mean BRAF-V600E mutation. In the second paragraph of the results, it is not clear whether this is all BRAF mutations (all variants and fusion) or just BRAF-V600E. This should be clarified here and elsewhere.

The one ATC with no cancer alteration is fascinating, but it not clear in the Results whether this particular tumor was analyzed by WGS or WES. Moreover, in Figure 1 there appears to be two such ATCs without any drivers. Can this be expanded and clarified.

In Figure 1 it is difficult to visualize the cases with gene fusions. Maybe they could break out the fusion cases in a separate section similar to how they did it for TERT and ATRX.

In summary, this is an excellent study that builds on the work started by the TCGA study of papillary carcinoma and other subsequent studies and provides critical insights into the development of advanced and lethal forms of thyroid cancer.

Response to reviewer(s)' comments

Reviewer #1 (Remarks to the Author):

In the present manuscript the Authors investigated both mutational landscape and transcriptome of anaplastic thyroid cancer (ATC) and advanced differentiated thyroid cancer (DTC) (total 113 patients). By doing this they identified a novel ATC-like subtype of thyroid cancer. Moreover, they observed that CDKN2A losses to be associated with poor survival and upregulation of genes coding for PD-L1 and PD-L2. Finally, the Authors reports the activation of potentially targetable VEGF-notch and JAK-STAT signaling pathways in ATC.

The topic is certainly of interest, even if the DNA sequencing data are not novel as also stated by the Authors. Some of the results might be relevant to improve our knowledge of molecular mechanisms involved in thyroid cancer development and progression. There are also a number of interesting preliminary insights about potential future targeted therapies. The manuscript is generally well written and the bioinformatics analysis is well performed. However, I do have some major concerns, especially about the non-homogeneity of the sequencing methods used and the lacking of functional validation.

Thank you for the reviewer's kind comments. We really appreciate the reviewer's crucial opinion and we value them very much. We tried to deliver our point on this manuscript by answering all the comments thoroughly.

Major comments

General: some assertions in the Abstract and in the Discussion should be lowered down as there is no functional validation shown. For instance, from presented data, it is difficult to say "TERT induces early progression of DTC".

[Response]

We appreciate the comment. As the reviewer recommended, we changed some descriptions in Abstract and Discussion sections as follows.

[Revised] page 2 line 52-55 in Abstract section

TERT, AKT1, PIK3CA, and EIF1AX were frequently co-mutated with main driver genes (BRAFV^{600E} and RAS) in advanced DTCs as well as ATC, but tumor suppressor (e.g., TP53 and CDKN2A) alterations were predominantly found in ATC.

[Revised] page 13 line 288-290 in Discussion section

Therefore, this result again underscores the importance of TERT in metastatic, invasive, and early aggressive characteristics of DTC, rather than anaplastic transformation of TC.

[Revised] page 13 line 299-300 in Discussion section

We also confirmed that the prevalence of second- and third-hit in oncogenes and TSGs were increased in ATC.

[Revised] page 14 line 311-314 in Discussion section

Altogether, our results exhibited the potential contribution of TERT and diverse oncogenes (AKT1/PIK3CA and EIF1AX) in early progression of DTC, and higher relation of TSGs (e.g., TP53 and CDKN2A) in anaplastic change of TC. Notably, CDKN2A loss may be a strong prognostic factor for patients with advanced DTCs as well as ATC.

Methods:

-the tumour DNA-samples were investigated by using different methods: 17 by WGS, 8 by WES and 88 by targeted sequencing (panel of 57 genes, 13 regions for fusion rearrangements and one region for TERT promoter mutations), sometimes with different sequencing depth. The Authors should clearly state to what extent this variability might affect their results. Moreover, they should specify how did they chose the genes contained in the panel for targeted sequencing.

[Response]

Thank you for the important comments. We have performed preliminary analysis by WGS (13 ATCs and 3 focal ATC/PDTCs) and WES (9 wiFTCs) and expanded the study with the additional samples by

targeted sequencing. For targeted sequencing, we selected genes for panel from five previous studies (TCGA, 2014; Yoo et al, 2016; Landa et al; 2016; Stransky et al, 2014; Costa et al, 2015) and identified in preliminary analysis of WGS/WES from this study. Moreover, four genes (*STARD9*, *HUWE1*, *BAZ2B*, and *MCM6*) which were discovered in our unpublished work about distant metastasis of FTC were included. We have specified and referenced the source of selected genes in methods section and supplement table 6 as a footnote.

We agree that different sequencing methods could lead to unwanted bias and WGS could not efficiently detect low frequency somatic mutations due to their poor sequencing depth relative to WES and targeted sequencing. However, we achieved deep sequencing depth of WGS (on average, 72.23X for tumors), and we think it could minimize the bias. To validate the difference between WGS and targeted sequencing, we also performed targeted sequencing on 16 ATCs which were analyzed by WGS. The results between WGS and targeted sequencing were compared, and they were highly concordant (91.89%). We stated and presented this data in Results section and as supplementary table 1.

On the other hand, one wiFTC tumor was analyzed by WGS as well as WES/RNA-seq. This case was initially analyzed by WES/RNA-seq same to the other eight wiFTCs. In the analysis of RNA seq, we found the overexpression of *TERT*, but we could not find any fusion or promoter mutation of *TERT* gene. Therefore, WGS was applied to identify a potential *TERT* rearrangement located in intergenic region as previous studies (Valentijn. et al 2015; Peifer et al 2015).

We described more clearly about the DNA sequencing methods applied in the present study as follows:

[Revised] page 5, line 91-98 in Results section

We have preliminarily analyzed 13 ATCs, 3 focal ATC/poorly differentiated TCs (PDTCs), and 9 widely invasive follicular TCs (wiFTCs) by whole-genome sequencing (WGS) or whole-exome sequencing (WES), and extended the dataset with 88 additional samples using targeted sequencing. In total, 113 advanced TCs, including 27 ATCs, 15 PDTCs, 28 focal ATC/PDTCs, 12 wiFTCs, and 31 metastatic papillary TCs (PTCs) were investigated for mutational profiling. Targeted sequencing was also performed on 13 ATCs and 3 focal ATC/PDTCs which were analyzed by WGS and the concordance

rate between two methods was 91.89% (Supplementary Table 1).

[Revised] page 19, line 421-424 in Methods section

The target genes were selected by our preliminary findings from WGS/WES and the previously reported genes in TC^{1, 2, 8, 49, 50}. Moreover, four genes (STARD9, HUWE1, BAZ2B, and MCM6) which were discovered in our unpublished work about distant metastasis of FTC were included.

[Revised] Supplementary Table 6

Supplementary Table 6. The list of genes captured by custom probes.

Target	Gene symbols
Small size mutations	AKT1, AKT3, ARID1A, ARID1B, ARID2, ARID5B, ATM, ATRX, BAZ2B ^a , BRAF, CDKN2A, CHEK2, CTNNA2, DICER1, EIF1AX, EZH1, HRAS, HUWE1 ^a , IDH1, KMT2A, KMT2C, KMT2D, KRAS, LATS1, LATS2, MCM6 ^a , MEN1, MLH1, MSH2, MSH6, MTOR, NF1, NF2, NFE2L2, NRAS, PBRM1, PIK3C2G, PIK3C3, PIK3CA, PIK3CG, PIK3R1, PIK3R2, PPM1D, PTEN, RB1, SETD2, SMARCB1, SOS1, SPOP, STARD9 ^a , STK11, TET1, TP53, TSC1, TSC2, TSHR, U2AF1
Fusion genes	ALK, B4GALNT3, BRD4, FGFR2, FGFR3, MET, NTRK1, NTRK3, NUTM1, PAX8, RET, THADA
Promoter mutations	TERT

^aGenes from our unpublished work about distant metastasis of FTC.

[Reference]

1. Cancer Genome Atlas Research Network. Integrated genomic characterization of papillary thyroid carcinoma. Cell 159, 676-690 (2014).
2. Yoo SK, et al. Comprehensive Analysis of the Transcriptional and Mutational Landscape of Follicular and Papillary Thyroid Cancers. PLoS Genet 12, e1006239 (2016).
8. Landa I, et al. Genomic and transcriptomic hallmarks of poorly differentiated and anaplastic thyroid cancers. J Clin Invest 126, 1052-1066 (2016).
49. Stransky N, Cerami E, Schalm S, Kim JL, Lengauer C. The landscape of kinase fusions in cancer. Nat Commun 5, 4846 (2014).
50. Costa V, et al. New somatic mutations and WNK1-B4GALNT3 gene fusion in papillary thyroid

carcinoma. Oncotarget 6, 11242-11251 (2015).

[Revised] page 7, line 149-153 in Results section

We found remarkably elevated TERT expression in wiFTC with PDE8B-TERT and PTC with MTMR12-TERT from TCGA (Fig. 2h and Supplementary Fig. 4). Meanwhile, one wiFTC showed increased expression of TERT without fusion or promoter mutations, hence we performed WGS to identify structural variation adjacent to TERT as previous reports^{18,19}. As a result, an inter-chromosomal translocation, t(2;5)(2q;5p), at 21 kilobases upstream from TERT was discovered (Supplementary Fig. 5).

[Reference]

18. Valentijn LJ, et al. TERT rearrangements are frequent in neuroblastoma and identify aggressive tumors. Nat Genet 47, 1411-1414 (2015).

19. Peifer M, et al. Telomerase activation by genomic rearrangements in high-risk neuroblastoma. Nature 526, 700-704 (2015).

- it is not entirely clear how was selected the series for transcriptomic profiling and how the data were analysed. At line 344-345 “RNA-seq data of 162 patients with DTC from our previous study were analysed with those of advanced TCs”. Do the Authors mean that the data from the previous 162 DTC were used as reference for the new series of 113 advanced TCs

[Response]

We apologize for the unclear description. As the reviewer mentioned, 162 DTCs were reference dataset for new series of RNA sequenced 25 advanced TCs. We added a description about the 162 samples using as reference dataset in Results and Methods sections for clear description

[Revised] page 9, line 217-218 in Results section

Transcriptome of 13 ATCs, 3 focal ATC/PDTCs, and 9 wiFTCs which were sequenced by WGS or WES were profiled by RNA-seq, then compared with data from 162 DTCs of our previous study².

[Revised] page 17, line 379-380 in Methods section

For transcriptomic profiling, RNA-seq data of 162 patients with DTC from our previous study² were applied as reference dataset for newly sequenced 25 samples.

- matched normal DNA was not analysed for the targeted sequenced tumors, which are actually the majority. This should be better specified in the Methods section. It can drive to some bias. For instance, as the Authors write in the results section, it is also possible that mutations in some genes such as KMT2D, ATM, CHEK2, NF1, NF2, MEN1 might be rare germline mutations. This may be true for TP53 gene as well. This might be a major issue and should be clearly stated and discussed.

[Response]

Thank you for the insightful comments. In Methods section, we specified that targeted sequencing was performed without matched normal sample. We also agree that some TP53 mutations might be rare germline mutation. Therefore, we confirmed that two TP53 mutations (E11Q and R49H) are germline mutation. For three samples (T19, T21, and T72), there were matched normal samples which were sequenced by next-generation sequencing, hence we provided IGV images of tumor/normal samples. For T22 which were analyzed by targeted sequencing, we confirmed germline mutation by Sanger sequencing of normal tissue. In case of T108, due to the absence of normal sample, we were not able to verify E11Q as germline mutation. However, it would be more reasonable to classify this mutation in T108 as germline, because minor allele frequency of E11Q in Korean population is quite high, which is 0.3% (Kwak et al 2017). We described this result in Results section.

[Revised] page 19, line 424-425, Methods section

The average sequencing depth of 421.54X were achieved and matched normal samples were not included.

[Revised] page 6, line 122-124, Results section

Notably, we confirmed two types of germline TP53 mutation (E11Q and R49H) in ATC (11.11%), wiFTC (8.33%), and metastatic PTC (3.23%; Supplementary Fig. 2).

[Revised] Supplementary Figure 2

Supplemental Fig. 2. Germline *TP53* mutations in study subjects.

[Reference]

Kwak SH, Chae J and Choi S et al., Findings of a 1303 Korean whole-exome sequencing study. *Exp Mol Med* 2017 Jul 14;49(7):e356

- the relationship between presence of CDKN2A loss and short survival looks noteworthy from the survival curves (Fig 3g-h, n=27 ATC and n=86 advanced DTC). However, it is not enough to conclude that CDKN2A loss might represent a prognostic factor. A multivariate regression analysis including other potential prognostic factors (e.g. presence of distant metastasis at diagnosis? Histopathological parameters? Etc) should be added. The Hazard Ratio and the 95% confidence interval must also be included. Importantly, the effect of CDKN2A loss on survival should be validated on an independent cohort of ATC should be performed.

[Response]

Thank you for the valuable comments. As the reviewer suggested, we performed Cox proportional hazards regression analysis to adjust potential prognostic factors (age, sex, distant metastasis, and tumor origin). After adjustment, we found that hazard ratios (HRs) for ATC and advanced DTCs were still significant as supplementary table 3 shown.

[Revised] Supplementary Table 3

Supplementary Table 3. Hazard ratios (HR) of CDKN2A loss for death in ATC and advanced DTCs.

Model	All (n=113)		ATC (n=27)		Advanced DTCs (n=86)	
	HR (95% CI)	P	HR (95% CI)	P	HR (95% CI)	P
Model 1	11.03 (4.97-24.45)	<0.001	2.95 (1.08-8.04)	0.034	31.36 (7.71-127.60)	<0.001
Model 2	13.59 (5.54-33.37)	<0.001	4.47 (1.33-15.01)	0.016	21.48 (4.56-101.14)	<0.001
Model 3	10.56 (4.29-25.96)	<0.001	3.90 (1.10-13.78)	0.035	15.00 (3.15-71.46)	<0.001
Model 4	9.61 (3.73-24.79)	<0.001	6.67 (1.34-33.12)	0.02	9.88 (1.97-49.57)	<0.001

Model 1. Unadjusted.

Model 2. Adjusted for age at surgery for analyzed tissue and sex.

Model 3. Adjusted for age at surgery for analyzed tissue, sex, and distant metastasis.

Model 4. Adjusted for age at surgery for analyzed tissue, sex, distant metastasis, and tumor origin.

[Revised] page 9, line 196-202 in Results section

Intriguingly, thyroid differentiation score (TDS) in ATCs with CDKN2A loss was significantly lower than those with wild-type (P < 0.001; Fig. 3e). We also found that CDKN2A loss was significantly associated with increased disease-specific mortality in patients with ATC and advanced DTCs (P =

0.03 and < 0.001 for each; Fig. 3f). The hazard ratios (HRs) were 6.67 (95% confidence interval [CI], 1.34-33.12) and 9.88 (95% CI, 1.97-49.57), respectively, after adjustments for the age at surgery, sex, distant metastasis, and tumor origin (Supplementary Table 3).

- **CDKN2A losses (or mutations or promoter hypermethylation) are frequent in other types of tumours and have been deeply investigated. Indeed, it should be validated in ATC at the protein level by investigating p16 expression. CDKN2A losses also has been linked to response to CDK4/6 inhibitors (ex. Elvin et al 2017). This should be added.**

- **Moreover, loss of CDKN2A has been previously linked to resistance to vemurafenib treatment in metastatic BRAF-PTC cells (Duquette et al 2015). This should be considered and at least discussed.**

[Response]

We validated the relationship between *CDKN2A* loss and patient outcome by p16 immunohistochemistry (IHC) using tissue microarray (TMA) of 57 ATC and advanced DTC samples. Among them, 42 samples were included in main sequencing analysis and we found that all samples (5/5) with *CDKN2A* loss were p16-negative. We showed the relationship between *CDKN2A* loss and p16 expression in supplementary figure 14 with representative images of sequencing and IHC results. With 57 samples with p16 IHC, we found that p16 expression also shows the association with disease-specific mortality of patients with ATC and advanced DTCs ($P = 0.03$ for both; Fig. 3g). Similar results were observed when we sub-analyzed using the 15 un-overwrapped samples. We also performed Cox proportional hazards regression analysis, but it was only significant after adjusting the age at surgery and sex (Supplementary Table 4). This might be due to smaller sample size than main analysis. Regarding the association between *CDKN2A* loss and response to CDK4/6 inhibitors and resistance to vemurafenib treatment, we totally agree with the reviewer's comments; we discussed this issue in discussion section.. .

Figure. The effect of p16 expression on disease-specific survival patients with advanced TC from independent cohort of 15 samples (This figure was not include in the manuscript).

[Revised] Supplementary Figure 14

Supplementary Figure 14. p16 immunohistochemistry using tissue microarray analysis. a) The representative images of p16-negative (upper) and p16-positive (lower) results. b) The relationship between *CDKN2A* loss and p16 expression.

[Revised] Figure 3g

Figure 3. g) The effect of p16 expression on disease-specific survival in ATC and advanced DTCs.

[Revised] Supplementary Table 4

Supplementary Table 4. Hazard ratios (HR) of p16 expression for death in ATC.

Model	All (n=57)		ATC (n=17)	
	HR (95% CI)	P	HR (95% CI)	P
Model 1	13.86 (1.83-105.20)	0.011	7.10 (0.890-56.04)	0.063
Model 2	10.67 (0.09-1.38)	0.023	35.25 (1.38-898.79)	0.031
Model 3	2.58 (0.25-26.22)	0.424	30.05 (0.93-973.91)	0.056
Model 4	1.13 (0.31-30.72)	0.333	5.17 (0.29-93.11)	0.266

Model 1. Unadjusted.

Model 2. Adjusted for age at surgery for analyzed tissue and sex.

Model 3. Adjusted for age at surgery for analyzed tissue, sex, and distant metastasis.

Model 4. Adjusted for age at surgery for analyzed tissue, sex, distant metastasis, and tumor origin.

[Revised] page 9, line 204-214 in Results section

We validated the relationship between CDKN2A loss and patient outcome by p16 immunohistochemistry (IHC) using tissue microarray (TMA) of 57 ATC and advanced DTC samples, of which 42 were sequenced samples (Supplementary Fig. 14a). All tissues (5/5) with CDKN2A loss, and 43.2% of CDKN2A wild-type (16/37) were p16-negative (Supplementary Fig. 14b). p16

expression displayed the association between poor disease-specific survival in patients with ATC and advanced DTCs ($P = 0.03$ for both; Fig. 3g). In ATC, p16-negative status increased the risk of disease-specific mortality (HR, 35.25; 95% CI, 1.38-898.79) after adjustments for the age at surgery and sex, although the statistical significance was lost without adjustment for covariates or after additional adjustments for distant metastasis and tumor origin (Supplementary Table 4). In advanced DTCs, the hazard ratio was not calculated since p16-positive patients were all censored.

[Revised] page 15, line 335-338 in Discussion section

In addition to the aforementioned pathways, previous reports showed that cell lines with CDKN2A/p16 loss is linked to response to CDK4/6 inhibitors such as palbociclib^{33, 34, 35} and to resistance to BRAF^{V600E} selective inhibitor, vemurafenib, in metastatic BRAF^{V600E} PTC cells³⁶.

[Reference]

33. Elvin JA, et al. Clinical Benefit in Response to Palbociclib Treatment in Refractory Uterine Leiomyosarcomas with a Common CDKN2A Alteration. *Oncologist* 22, 416-421 (2017).

34. Konecny GE, et al. Expression of p16 and retinoblastoma determines response to CDK4/6 inhibition in ovarian cancer. *Clin Cancer Res* 17, 1591-1602 (2011).

35. Wiedemeyer WR, et al. Pattern of retinoblastoma pathway inactivation dictates response to CDK4/6 inhibition in GBM. *Proc Natl Acad Sci U S A* 107, 11501-11506 (2010).

36. Duquette M, et al. Metastasis-associated MCL1 and P16 copy number alterations dictate resistance to vemurafenib in a BRAFV600E patient-derived papillary thyroid carcinoma preclinical model. *Oncotarget* 6, 42445-42467 (2015).

Results:

-did the Authors find any difference in terms of genomic alterations and/or transcriptome profile among samples coming from primary tumours, local recurrences or distant metastasis?

[Response]

We checked the potential difference between primary, distant metastatic, and locally recurred tumors

in transcriptome level by principal component analysis. However, there was no correlation derived by the source of sequenced tissue. Due to the limitation of the length of manuscript, it was not included in manuscript. However, if the reviewer recommends to add, we are willing to add this points in Discussion with supplementary results.

Figure. Principal component analysis of 13 ATC and 3 focal ATC/PDTC samples using RNA expression (Data not shown in main text). Shape and color represent the analyzed tissue type and genetic alteration (This figure was not include in the manuscript).

In case of genetic alteration, we could not compare the difference between primary, local recurrence or distant metastatic tumors since we did not sequence multiple tumors from single patient. However, recent report (Dong et al, 2018) displayed that subclonal heterogeneity in thyroid cancer is low. Therefore, we might suppose that the genetic basis of distant or lymph node metastasis tissue would not so significantly differ from that of primary tumor.

[Reference]

Dong W, et al. Clonal evolution analysis of paired anaplastic and well-differentiated thyroid carcinomas reveals shared common ancestor. *Genes Chromosomes Cancer* 57, 645-652 (2018).

- lines 124-125: findings about mutations in ATRX, NFE2L2 and SPOP should be shortly presented in the text.

[Response]

We described three genes and *RET* fusion in Results section.

[Revised] page 5, line 104-105 in Results section

RET fusions (*CCDC6-RET* and *NCOA4-RET*) were discovered in PDTC, focal ATC/PDTC, and metastatic PTC.

[Revised] page 5, line 105-107 in Results section

We also found *NFE2L2* mutation which is frequently altered in lung squamous cell carcinoma and recently identified in TC as fusion driver^{10, 11}.

[Revised] page 5, line 111-112 in Results section

In metastatic PTC, we also discovered *SPOP*^{P94R} which is repeatedly reported in various types of TC^{2, 12, 13}.

[Revised] page 6, line 118-119 in Results section

We also identified *ATRX* mutations in ATC, focal ATC/PDTC, wiFTC, and metastatic PTC.

- line 241-242 and 243-244: from text and Fig 5d/5e, it is not fully clear in how many ATC samples were the expression of the genes coding for PD-L1 (CD274) and PD-L2 (PDCD1LG2) up-regulated.

[Response]

We modified the box plots to recognize how many ATCs accompany the overexpression of *CD274* (PD-L1) and *PDCD1LG2* (PD-L2) (Fig 5e).

[Revised] Figure 5e

Figure 5. e) The expression levels of *CD274* and *PDCD1LG2* in various types of TC.

Moreover, I cannot see in Fig 5e the relationship between this overexpression and survival.

[Response]

We intended to describe the relationship between the *CDKN2A* loss and survival based on Figure 3 rather than overexpression of PD-L1/PD-L2 and survival. However, the sentence “which showed the poorest thyroid differentiation and shorter disease-specific survival” could lead to misunderstanding of the result, as the reviewer pointed out, so we removed the sentence from the original description.

Finally, in the present study, there is no direct evidence to explain the connection between PD-L1 (*CD274*) and PD-L2 (*PDCD1LG2*) and *CDKN2A* losses.

[Response]

We agree that the underlying mechanism of *CDKN2A* loss and the up-regulation of PD-L1/PD-L2 is not elucidated in this study. In original manuscript, we have stated this issue in the Discussion section as “The underlying reason of the increased expression of *CD274* and *PDCD1LG2* in ATC with *CDKN2A* loss is not unveiled in this study, but their potential relationship in non-small cell lung cancer were previously reported^{41, 42}.” and cited similar reports. To add the statistical significance of their relationship, we performed differentially expressed gene analysis and specified the Log₂fold-change in Results section.

[Revised] page 12, line 275-276 in Results section

Interestingly, the up-regulation of these genes were found in ATCs with CDKN2A loss (Log_2 fold-change [FC] = 2.38 and 2.85 for each; Fig. 5e).

[Reference]

41. Zhang Y, Marin-Acevedo JA, Azzouqa AG, Manochakian R, Lou Y. Association of CDKN2A gene alteration with high expression of PD-L1. *Journal of Clinical Oncology* 36, 9102-9102 (2018).
42. Sterlacci W, Fiegl M, Droeser RA, Tzankov A. Expression of PD-L1 Identifies a Subgroup of More Aggressive Non-Small Cell Carcinomas of the Lung. *Pathobiology* 83, 267-275 (2016).

-the Authors observed that genes involved in VEGF-notch and JAK-STAT signalling pathways are overexpressed in BRAF-positive ATC. In order to propose these potentially targetable pathways for developing future treatment options for ATC, specific functional experiments in ATC cell lines should be performed.

[Response]

We really appreciate the reviewer's important comments. As the reviewer recommended, we checked effects of inhibition of the VEGF-Notch signaling pathway in *BRAF*-positive ATC or the JAK-STAT signaling pathway in *RAS*-positive ATC. In CAL62 cells, the *RAS*-positive ATC cells, a JAK inhibitor, ruxolitinib could decrease the expression of *BCL2L1*, *SOCS3*, and *MYC*, the downstream molecules of JAK-STAT pathway, resulting decreased cellular proliferation. However, we failed to demonstrate effects of inhibition of the VEGF-Notch signaling pathway by DAPT, a Notch signaling inhibitor, in *BRAF*-positive ATC cells. The effect of VEGF-Notch signaling inhibition may not have been shown due to cross-regulation of other activated signal pathways. Considering these points, it could not be regarded as the main signaling pathway to be the therapeutic target in *BRAF*-positive ATC. We deleted the previous Fig. 5b and the corresponding part, and mentioned these points in the Discussion section. Furthermore, we only kept JAK-STAT signaling pathway in Abstract section.

[Revised] Figure 5c

Figure 5 c). Quantitative reverse transcription polymerase chain reaction measurement of expression of JAK-STAT signaling pathway genes

[Revised] Figure 5d

Figure 5 d). cell viabilities analyzed by cell counting kt-8 assay, in CAL72 cells following treatment with ruxolitinib (1, 10, 20, 25 μM). Ctrl denotes control. *P < 0.05 versus controls.

[Revised] page 11, line 264-269 in Results section

In order to evaluate the potential ability of activated pathways in ATC as druggable targets, we performed functional in vitro experiments with ATC cell lines. In BRAFV600E-positive ATC cell lines, we were unable to demonstrate effects of inhibition of VEGF-Notch signaling. Whereas, in CAL62, the RAS-positive ATC cells, JAK inhibition with ruxolitinib decreased the

expression of SOCS3, BCL2L1, and MYC which are the downstream molecules of JAK-STAT pathway (Fig. 5c), and then we confirmed reduced cellular proliferation (Fig. 5d).

[Revised] page 15, line 331-335 in Discussion section

Although we demonstrated that the cell viability was regulated by inhibition of the activated JAK-STAT signaling in RAS-positive ATC cell line in vitro, inhibition of Notch signaling did not affect BRAF^{V600E}-positive ATC cell lines. It is possible that the effect might not have been shown, since there would be other upstream pathways that regulate VEGF-Notch signaling³².

[Reference]

32. Song YS, et al. Aberrant Thyroid-Stimulating Hormone Receptor Signaling Increases VEGF-A and CXCL8 Secretion of Thyroid Cancer Cells, Contributing to Angiogenesis and Tumor Growth. Clin Cancer Res 25, 414-425 (2019).

[Revised] page 3, line 58-59 in Abstract section

Furthermore, the activation of JAK-STAT signaling pathway could be potential druggable target in RAS-positive ATC.

Minor comments:

Introduction, line 79: “to discover favourable targets for....” Should be modified into “to discover molecular mechanisms potentially involved in tumour progression and/or as targets for treatment”.

[Revised]

Thank you for the kind suggestion. We modified the sentence as the reviewer suggested.

[Revised] page 4, line 80-82 in Introduction section

Thus, the need for further transcriptomic analysis of ATC and advanced DTCs is increased to discover molecular mechanisms potentially involved in tumor progression and targets for treatment.

Results, lines 88-89: the number of samples investigated by WGS, WES and targeted sequencing needs to be added here.

[Response]

We described the number of samples which were analyzed by WGS, WES, and targeted sequencing.

[Revised] page 5, line 91-94 in Results section

We have preliminarily analyzed 13 ATCs, 3 focal ATC/poorly differentiated TCs (PDTCs), and 9 widely invasive follicular TCs (wiFTCs) by whole-genome sequencing (WGS) or whole-exome sequencing (WES), and extended the dataset with 88 additional samples using targeted sequencing.

Results, line 180: how was exactly the “poorest thyroid differentiation” evaluated?

[Response]

It was assessed by thyroid differentiation score (TDS). In revised manuscript, we have stated it as “thyroid differentiation score” rather than “thyroid differentiation”.

[Revised] page 9, line 196-197 in Results section

Intriguingly, thyroid differentiation score (TDS) in ATCs with CDKN2A loss was significantly lower than

ATC with CDKN2A wild-type ($P < 0.001$; Fig. 3e).

Legend to the Figures: TDS should be spelled.

[Response]

We have spelled TDS and BRS as Thyroid differentiation score $BRAF^{V600E}$ -RAS score in Figure legends.

[Revised] Figure legends.

Figure 3. e) The effect of CDKN2A loss on thyroid differentiation score (TDS) in ATC.

[Revised] Figure legends

Figure 4. b) The results of $BRAF^{V600E}$ -RAS score (BRS) analysis were represented by box plots.

We really appreciate the reviewer's kind and thoughtful comments.

Reviewer #2 (Remarks to the Author):

The study by Yoo et al describes a genomic analysis of several pathological types of follicular cell thyroid cancers, including the most lethal type, anaplastic thyroid carcinoma. The cohort for DNA sequencing consisted of primary tumors from thyroidectomies of 27 anaplastic cancers and 86 clinically advanced differentiated thyroid cancers that included 15 poorly differentiated carcinomas, 28 cases with focal anaplastic or poorly differentiated carcinoma, 12 widely invasive follicular carcinoma and 31 papillary carcinomas associated with metastatic disease. In addition, RNA sequencing data from a previous study was integrated in the analysis.

We are very grateful for the valuable comments and suggestions. We tried our best to answer the questions and suggestions of reviewer, which we address below.

The resulting analysis is highly data rich and well presented. The information presented definitely advances the field, but there are some reasons for concern that need to be addressed.

The major criticism of the study relates to study design, specifically the variable use of WGS, WES and targeted sequencing across the different pathologic subtypes. For example, according to the legend in Figure 1, WGS was applied to ATC and a few “focal ATC/PDTC” and none of the PDTC or metastatic PTCs. WES was only applied to “WiFTCs” and, by default, targeted sequencing was only applied to PDTC and metastatic PTC. This design is clearly suboptimal as it introduces potential bias on the data because WGS will naturally find more variants than WES and targeted sequencing. In other words, they mixed the depth and range of sequencing across the different tumor histologies. This was likely not by design and should be explained how it came to be. It is not clear that this was normalized in the analysis, if possible. Was anything done to control or adjust for this? At least this should be discussed and highlighted as a potential limitation and source of bias.

Thank you for the kind comments. We have started preliminary analysis by WGS (13 ATCs and 3 focal ATC/PDTCs) and WES (9 wiFTCs) and expanded the study with the additional samples by targeted sequencing. As the reviewer mentioned, WGS could result in more variants than WES and targeted sequencing. However, variant analysis regarding oncogenes and tumor suppressors were restricted to exonic and splicing regions, hence we believe that the bias between WGS and WES is not high. Furthermore, our targeted sequencing panel was designed based on preliminary findings of WGS/WES and five previous study related to thyroid cancers (TCGA, 2014; Yoo et al, 2016; Landa et al; 2016; Stransky et al, 2014; Costa et al, 2015). Moreover, four genes (*STARD9*, *HUWE1*, *BAZ2B*, and *MCM6*) which were discovered in our unpublished work about distant metastasis of FTC were included. We are quite confident that our effort would also minimize the bias derived by different sequencing methods. We described the targeted sequencing design in Methods section.

Moreover, only WGS and WES were performed with matched normal samples, therefore targeted sequencing data would harbor potential bias regarding germline mutation. In Methods section, we specified that targeted sequencing was performed without matched normal sample. Moreover, we have presented selected genes which were confirmed to have somatic mutations from WGS/WES analysis in Figure 1 as main finding. We have mentioned this manner in Figure 1 legend, but also described it in method section to show the effort to adjust this bias. Also, we have performed targeted sequencing on 16 ATCs which were analyzed by WGS, then compared the results between WGS and targeted sequencing, and they were highly concordant (91.89%). We also presented this data as supplementary table 1.

In contrast to genetic alterations in exonic region, those located in intronic or intergenic regions were not able to identify by WES or targeted sequencing. Therefore, one of the major finding in our study, *TERT* rearrangements, might be undetected in the majority of study subjects since only 17 samples were sequenced by WGS. We described the possibility that the prevalence of *TERT* rearrangement in advanced thyroid cancer may be higher than we reported.

[Revised] page 19, line 421-424 in Methods section

The target genes were selected by our preliminary findings from WGS/WES and the previously

reported genes in TC^{1, 2, 8, 49, 50}. Moreover, four genes (*STARD9*, *HUWE1*, *BAZ2B*, and *MCM6*) which were discovered in our unpublished work about distant metastasis of FTC were included.

[Reference]

1. Cancer Genome Atlas Research Network. Integrated genomic characterization of papillary thyroid carcinoma. *Cell* 159, 676-690 (2014).

2. Yoo SK, et al. Comprehensive Analysis of the Transcriptional and Mutational Landscape of Follicular and Papillary Thyroid Cancers. *PLoS Genet* 12, e1006239 (2016).

8. Landa I, et al. Genomic and transcriptomic hallmarks of poorly differentiated and anaplastic thyroid cancers. *J Clin Invest* 126, 1052-1066 (2016).

49. Stransky N, Cerami E, Schalm S, Kim JL, Lengauer C. The landscape of kinase fusions in cancer. *Nat Commun* 5, 4846 (2014).

50. Costa V, et al. New somatic mutations and WNK1-B4GALNT3 gene fusion in papillary thyroid carcinoma. *Oncotarget* 6, 11242-11251 (2015).

[Revised] page 19, line 424-425, Methods section

The average sequencing depth of 421.54X were achieved and matched normal samples were not included.

[Revised] page 20, line 455-458 in Methods section

Then, we separated genes into two groups to avoid potential bias derived from the absence/presence of matched normal samples as follows: 1) genes that were confirmed to have somatic mutations from WGS/WES analysis (Fig. 1); 2) genes that were highly suspected to have germline mutations (Supplementary Fig. 1).

[Revised] page 13, line 295-298 in Discussion section

Although two TERT rearrangements across 113 TCs were found, there might be more tumors with these alterations since WES and targeted sequencing approach which were performed on most of study subjects (85.84%) did not cover intronic and intergenic regions

Beyond the study design, I have several minor comments.

They use the BRAF-like and RAS-like distinction first developed by the TCGA study. They should use the more specific term, BRAF-V600E-like, because not all BRAF mutations have a BRAF-V600E-like phenotype.

[Response]

We have changed *BRAF*-like to *BRAF*^{V600E}-like in the manuscript.

The definition of focal ATC/PDCA is not clear and this should be defined using pathological terms.

[Response]

We described the definition of focal ATC/PDTC in a more detailed manner in methods section.

[Revised] page 18, line 388-397 in Methods section

As there is no definitive established pathologic definition for this, the definitions adopted in this study were based on the experience of clinicians and pathologists. The ATC component in tumors mixed with DTC was defined based on the following features: the nuclei without the characteristic features of DTC and showing a greater ratio of nucleus/cytoplasm, nuclear pleomorphism other than the features of DTC, and a more solid growth pattern with or without p53 expression. PDTC was defined on the basis of the Turin proposal for the use of uniform diagnostic criteria³⁹, and was confirmed if showing a solid, trabecular, or insular growth pattern with the absence of conventional nuclear features of papillary carcinoma, and the presence of at least one of the following features: tumor necrosis, mitotic count $\geq 3/10$ high-power field, or convoluted nuclei.

The metastatic nature of the PTCs should be highlighted in the main text. Where these patients with distant metastatic disease or regional lymph nodes metastases. They should also make it very clear that they studied the primary tumors, not the metastatic tumors. It was confusing at first reading and I recognize there is data in Figure 1, but something more should be added in

the Results to help the reader understand what was studied.

[Response]

Thank you for kind comment. In this study, metastatic PTCs were tumors that all have distant metastasis to other organs. Also 77.42% of them (24/31) had lymph node metastasis. We described the metastatic status of PTC samples in methods section and modified figure 1 to illustrate them. Moreover, we collected and sequenced three types of tissues not only primary tumors: 1) primary tumor, 2) locally recurred or residual tumor, and 3) distant metastatic tumor. We stated this manner in Results section to give a clear description of the study design.

[Revised] Figure 1

Figure 1. The mutational landscape of ATC and advanced DTCs.

[Revised] page 5, line 98-99 in Results section

We collected tissues from primary (76/113), distant metastatic (19/113), and locally recurred or residual sites (18/113), respectively.

[Revised] page 18, line 398-399 in Methods section

All metastatic PTC accompanied distant metastasis in other organs and 77.42% of them (24/31) also had lymph node metastasis.

BRAF mutation is used in the paper when I think they mean BRAF-V600E mutation. In the second paragraph of the results, it is not clear whether this is all BRAF mutations (all variants and fusion) or just BRAF-V600E. This should be clarified here and elsewhere.

[Response]

We have changed *BRAF* mutation to *BRAF*^{V600E} mutation in the manuscript.

The one ATC with no cancer alteration is fascinating, but it not clear in the Results whether this particular tumor was analyzed by WGS or WES. Moreover, in Figure 1 there appears to be two such ATCs without any drivers. Can this be expanded and clarified.

[Response]

Thank you for the comment. There were two ATCs which did not have putative driver mutation and they were sequenced by WGS/targeted sequencing and targeted sequencing only. Because the sequencing methods which applied to two ATC without driver mutation were well recognizable in Figure 1, we described that RNA-seq was performed on 25 samples which were sequenced by WGS or WES.

[Revised] page 9, line 217-219 in Results section

Transcriptome of 13 ATCs, 3 focal ATC/PDTCs, and 9 wiFTCs which were sequenced by WGS or WES were profiled by RNA-seq, then compared with data from 162 DTCs of our previous study².

In Figure 1 it is difficult to visualize the cases with gene fusions. Maybe they could break out the fusion cases in a separate section similar to how they did it for TERT and ATRX.

[Response]

We modified Figure 1 and separated *RET* section in the middle of oncogene/TSG and *TERT/ATRX* sections.

[Revised] Figure 1

Figure 1. The mutational landscape of ATC and advanced DTCs.

In summary, this is an excellent study that builds on the work started by the TCGA study of papillary carcinoma and other subsequent studies and provides critical insights into the development of advanced and lethal forms of thyroid cancer.

We really appreciate the reviewer's kind and thoughtful comments.

REVIEWERS' COMMENTS:

Reviewer #1 (Remarks to the Author):

The Authors answered to all my questions and modified the text and Figures accordingly. By doing so, the clearness and the value of the current version of manuscript significantly increased. To the best of my knowledge, the statistical analysis is appropriate and the results are now more convincing.

I do not have any further concerns about the paper.